# Randomly Sampled Language Reasoning Problems Elucidate Limitations of In-Context Learning

## Abstract

While LLMs have revolutionized the field of machine learning due to their high performance on a strikingly wide range of problems, they are also known to hallucinate false answers and underperform on less canonical versions of the same tasks. There are several emerging theories of LLM performance, among them that LLMs lack world modeling ability, that they have an undesirable bias towards an autoregressive prior, and that they struggle on more novel problems. The existing literature on LLM input novelty has focused on tasks of relatively high complexity, studying perturbations of canonical but complex problems. In this paper, we attempt to minimize complexity in order to isolate novelty as a factor in LLM underperformance and investigate the power of in-context-learning. To this end, we consider an extremely simple domain: next token prediction on simple language tasks. The twist is that these language tasks are wholly unseen, as they are randomly drawn from a large, parsimoniously defined set of languages arising from simple grammar rules. This experimental setup allows us to evaluate ICL independently of models' parametric knowledge. We find that LLMs uniformly underperform n-gram models on this task, both when used as next token predictors and in chain-of-thought.

## 1 Introduction

One of the surprising capabilities of contemporary LLMs is their ability to perform in-context-learning (ICL), in which they learn by demonstration from provided input/output examplars to produce an appropriate output for a new but similarly structured input. This capability allows LLMs to generalize to tasks beyond interpolations of their training corpus, resulting in remarkably strong generalization capabilities. It is challenging to isolate the effects of ICL, as, given the large datasets LLMs are trained on, it is difficult to distinguish a model detecting novel patterns in examples from it being guided towards knowledge it gained during training. Akyürek et al. (2022) demonstrate that, in specially trained transformers, ICL is "true learning". However, it remains unclear to what degree the performance of foundation-model LLMs can be attributed to strong ICL capabilities.

The task best suited to isolating LLM ICL performance should have the following properties: (1) It should be a language completion task within the expressive power of an LLM. LLMs are capable of many tasks, but they are primarily models of language, and as such, language tasks are the most fair evaluations. (2) It should not require sophisticated world modeling to solve. This helps us eliminate a possible source of underperformance distinct from ICL capability. (3) It should be selected at random in an unbiased manner to reduce the effect of bias from the training corpus.

To satisfy these properties, we propose the following general approach. First we define a large, exhaustive, and parsimoniously-defined space of languages that represents all languages of a certain difficulty level. Then, we sample random languages from this space. By sampling randomly, we can guarantee no bias towards canonical languages that might share structure with common ones in the training dataset. In this work, we use languages recognized by 3-state DFAs as these are the lowest nontrivial difficulty level. [1] Finally, to ensure we are not measuring world modeling performance,

---

[1]This technique can be generalized to produce benchmarks of any difficulty level. For larger numbers of states, we would be able to guarantee that the majority of the exponentially large number of corresponding

we compare to $n$-GRAM baselines that are not capable of anything other than matching clusters of tokens.

Our results demonstrate that even for very simple language induction tasks that don't rely on world modeling or background knowledge, LLM ICL still underperforms simple language models when dealing with randomly sampled and likely unfamiliar problem instances. These results suggest that while LLMs can pick up some learning signal from examples in prompts, this in-context learning is not competitive with even very primitive forms of learning, and suggests that LLMs do not posses the ability to generalize to entirely novel language reasoning tasks.

In summary, we make the following contributions:

1. We introduce a benchmark for LLM ICL language reasoning evaluation, consisting of novel tasks.

2. We evaluate a suite of popular LLMs on instances of this benchmark and demonstrate that LLMs underperform compared to simple language model baselines.

3. We analyze the differences in behavior between these models, illustrating the influence of model size and finetuning on language problem ICL capacity.

## 2 RELATED WORK

LLMs are known to fail in many cases, with some suggesting that these failures are due to lack of a world model (Valmeekam et al., 2022) or "embers" of autoregression polluting non-autoregressive task performance (McCoy et al., 2023). Another theory is that of task novelty; that is, LLMs perform worse on tasks more dissimilar from those seen during training.

### 2.1 LANGUAGE UNDERSTANDING AND LLMS

LLMs can be quite adept at generating programs in general-purpose programming languages (Xu et al., 2022a). In contrast, adapting models to understand domain-specific languages (Mernik et al., 2005) introduces unique problems such as navigating idiosyncratic syntax and semantics and leveraging sparse sample language data. To address these challenges, researchers have considered how well general-purpose LLMs can use language reasoning skills to quickly understand rare or unseen DSLs with only a small set of exemplars (Joel et al., 2024). While most work in this vein focuses on semantic parsing for downstream applications (Lin et al., 2023), selecting exemplars (Zhao et al., 2021), and improving DSL recognition by leveraging more common languages (Bogin et al., 2023), experiments show strong baseline performance for LLM DSL recognition and parsing out-of-the-box (Wang et al., 2024). Some have suggested that indicate that LLMs may possess emergent language reasoning abilities (Millière, 2024).

Related lines of work are compositional generalization (Xu et al., 2022b), which assesses models' ability to organize known units into novel structures, and structural generalization (Yao & Koller, 2022), which assesses models' ability to recognize new structures. Yao & Koller (2022) show that

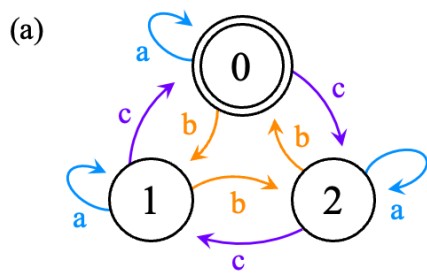

Figure 1: We sample randomly generated languages to test LLMs by sampling deterministic finite automata (DFAs). (a) The DFA shown here, modeling the sum modulo 3 operation (with `abc` representing 0, 1, and 2 respectively), can be used to accept or reject strings from a 3-character alphabet. Accepted strings belong to the grammar; rejected strings do not. We evaluate models on their ability to (b) act as a transducer, recognizing strings that belong to the grammar, and (c) generate new strings following the grammar.

___

languages do not lie in the training dataset by a pigeonhole argument; unfortunately this does not apply to the relatively small set of 3-state DFAs (there are only 78786). However, they still represent an set of tasks of a particular difficulty level not biased towards the canonical.

smaller language models like BART and T5 can struggle on these tasks, but to our knowledge there are not comprehensive experiments extending this line of work to LLMs.

## 2.2 REASONING WITH LLMS

Reasoning is one of many "emergent abilities" (Wei et al., 2022a) possibly possessed by LLMs (Huang & Chang, 2022), although the nonlinear dependence of such emergent abilities on model size is disputed (Schaeffer et al., 2024). The chain-of-thought prompting technique (Wei et al., 2022b) has inspired a number of approaches to encourage the latent reasoning ability of models (Yao et al., 2023; Besta et al., 2024; Kojima et al., 2022), including neuro-symbolic methods (Hua & Zhang, 2022; Weir et al., 2023; 2024). Building on this, other work considers how to optimize exemplars used for in-context learning (Dong et al., 2022) and chain-of-thought prompting, known as "rationale refinement" (Liu et al., 2021; Fu et al., 2022). Problem-decomposition is also shown to be effective (Zhou et al., 2022; Khot et al., 2022).

## 2.3 LLM REASONING EVALUATION

LLM reasoning abilities are often tested on natural language benchmarks and commonly seen problems like arithmetic (Cobbe et al., 2021; Amini et al., 2019; Hendrycks et al., 2021), commonsense reasoning (Bhargava & Ng, 2022), and other, sometimes generative, tasks (Lake & Baroni, 2018; Pasupat & Liang, 2015; Lin et al., 2019) and task collections (Srivastava et al., 2022). LLMs have been shown to lack sufficient reasoning capability across a range of tasks including multi-step planning and complex inference (Valmeekam et al., 2022). Fan et al. (2023) introduce an LLM reasoning benchmark on algorithmic problems through NP-hard complexity, and Hazra et al. (2024) show that LLMs struggle to complete simple 3SAT problems. Patel et al. (2021) demonstrate that much of LLM mathematical reasoning can be explained by shallow heuristics, Razeghi et al. (2022) similarly find that term frequency in training data impacts models' in-context learning ability, and (McCoy et al., 2023) theorizes that "embers" of autoregression are polluting non-autoregressive task performance.

The effect of novelty on performance has also been explored in prior work, generally via investigating perturbations of existing tasks, taking existing problems (that are often inherently quite complex, and are only easy because they are well known, e.g., addition of numbers or logical reasoning over natural language) and changing one small aspect of the problem (Wu et al., 2024; Saparov et al., 2023). LLM ICL itself has been studied in prior work, with Kossen et al. (2023) demonstrating that LLMs use labels provided in exemplars in naturalistic tasks and de Wynter (2025) using a mixture of simple and complex canonical formal language tasks to demonstrate that LLM ICL outperforms kNN baselines on these tasks but is brittle to minor changes in task performance. We also investigate LLM ICL but push both language simplicity and language unfamiliarity to their limits, by exploring simple languages recognized by randomly sampled DFAs. This enables us to best isolate the power of ICL in the language domain, where LLMs should perform best.

## 2.4 TRAINING TRANSFORMERS ON FORMAL LANGUAGES

A key assumption behind this work is that the tasks we are using to evaluate LLMs are solvable by LLMs. Vafa et al. (2025) frame world modeling (a statistical model inferring the true underlying causal graph behind the data being observed) as a latent DFA identification task, finding that transformers trained on DFA traces (of massive DFAs representing board games and city maps) do not reconstruct the underlying DFA. Other work also trains language models on formal languages (Butoi et al., 2024; Bhattamishra et al., 2023; Valvoda et al., 2022) and probabilistic formal languages (Borenstein et al., 2024). Akyürek et al. (2024) find that transformers trained on 4-12 state DFA transducer traces more effectively learn to in-context-learn regular languages than RNNs or $n$-GRAM models; Edelman et al. (2024) demonstrate similar results on small Markov Chains. Therefore, in this work, where we evaluate much larger LLMs on much simpler 3-state DFAs, we can be confident that underperformance relative to $n$-GRAMS is not linked to inherent transformer limitations and must be instead related somehow to specific properties of foundation models.

# 3 DFA REASONING TASKS

## 3.1 DFAS AND REGULAR LANGUAGES

The original Chomsky Hierarchy (Chomsky, 1959) separates language into four types (Figure 2). We focus on the task of understanding Type 3 languages, the simplest form of language in the hierarchy, that are recognized by a Deterministic Finite Automata (DFAs) whose outputs are boolean ($\{0, 1\}$). Examples of languages recognized by DFAs include simple ones like `binary strings with an even number of ones`, and even such examples as `numbers in base 10 divisible by 7`. Type 3 languages are also known as regular languages, which are recognized by regular expressions.

One simple metric of the difficulty of a regular language is the number of states in the corresponding DFA, i.e., the amount of working memory.[2] 2-state DFAs have the property that their set of states is no larger than the output set $\{0, 1\}$, and, therefore, do not have any hidden state. We thus explore 3-state DFAs, as this is the simplest nontrivial case.

## 3.2 LANGUAGE REASONING TASKS

We define a language reasoning task as a task corresponding to some latent language $\mathcal{L}$, where a set of positive/negative examples is provided to a model, with the goal being either classification of a new string in this language, or completion of an existing string to place it in the language.

Figure 2: An illustration of Chomsky's hierarchy of languages, ranging from Type 0 to Type 3, which are defined by what formal models can recognize their grammars. In this work, we focus on the simplest language type in the hierarchy, regular grammars, which are recognized by deterministic finite automata (DFAs).

### 3.2.1 SEQUENCE COMPLETION TASK

We first pose a *sequence completion* task, in which models must complete a sequence in a given DFA's language. This mirrors how foundation models are trained using masked language modeling, where data is presented in this format, with several *example sequences* in a given language followed by a *distinct prefix* that needs to be completed.

To generate test cases for this task given a DFA, we (1) sample 30 example sequences of length 10 that this DFA accepts, and then (2) sample a distinct prefix of length 5 that is not a prefix of any of our 30 example sequences, with the property that there exists some length-$\leq 5$ *completion* of this prefix that the DFA would accept. The task is to find a completion (not necessarily the same completion found in sampling) of this prefix of between 1 and 5 characters such that the DFA accepts the full sequence. For details on sampling, see Appendix A.2.

We evaluate models by (1) sampling a DFA, (2) sampling 30 problem instances at random (each of which contains 30 example sequences and a distinct prefix), and then (3) computing a binary prediction score (whether or not the predicted completion creates a valid string in the language) for each instance separately, then computing a correctness metric as a fraction. We then average this metric over several sampled DFAs to produce our accuracy score.

### 3.2.2 TRANSDUCER TASK

While the sequence completion task is the natural one that comes to mind as a basic language task, it has a difficulty-gap problem. Specifically, many DFAs, including the one shown in Figure 1, recognize languages that are particularly difficult to identify based on a set of examples, unless you build some kind of world model.[3] This is problematic as we would like to be able to assess

---

[2]There are other metrics of difficulty, but we choose number of states as it is highly parsimonious.

[3]The difficulty gap exists because a set of recognized sequences of length 10 gives no direct insight into intermediate states between the first and tenth token. As such, to be able to utilize this information for languages

the performance of language models at pattern recognition, independent of their world modeling abilities. To assess pattern recognition, we explore the Transducer task.

In this task, an input sequence is annotated with an output at each token, the final output is masked, and the masked output is predicted by a language model. E.g., given the language `even number of 'a' tokens` and the input `abcabcaabbccaa`, the annotated string (all that is provided to the model) is `a0b0c0a1b1c1a0a1b1b1c1c1a0a` and the output to predict is `1`. For each problem instance, we provide 30 symbols, and for the first 29, the corresponding transducer output.

This task is significantly more transparent than the sequence completion task as the model has access to intermediate outputs, an (imperfect) proxy for intermediate state.

### 3.3 BASELINES

To contextualize LLM accuracies, we provide several baselines with varying degrees of sophistication.

**Sequence Completion Task**  For the Sequence Completion task, we have four baselines.

- RANDOM$_S$ baseline: produce a random string of length 5 characters. While this might seem redundant as it should have a success rate of 50%, in practice our rejection sampling approach (see Appendix A.2) leads to a slight bias towards DFAs with more accept states. This baseline measures that bias.

- COMMON-SUFFIX$_S$ baseline: find the completion $s$ of length between 1 and 5 that maximizes (# of occurrences as a suffix $\times |s|$). This baseline does not take the distinct prefix into account whatsoever.

- $n$-GRAM$_S$ baseline: we take the last $n - 1$ characters of the distinct prefix and search to see if they appear in any of the example sequences at a position where the sequence following is an appropriate length to be a completion (at least 1 but at most 5). We then take a plurality vote among the completions and return this, breaking ties arbitrarily. If there are no matches, we return the result of $(n - 1)$-GRAM$_S$. Technically these cover more than $n$ characters, since the completion is often $> 1$ character long; for simplicity, however, we keep the naming consistent with the Transducer baselines. Despite the similarity between an $n$-GRAM and a DFA in terms of token-to-token transitions, $n$-GRAMS do not have access to DFA hidden state and thus cannot solve arbitrary DFA language problems, regardless of $n$.

- BRUTE-FORCE$_S$: take all possible DFAs with 3 states and 3 symbols. Filter for ones that accept all the example sequences. Then try all remaining DFAs on all $3^5$ possible 5-length completions and return the completion that the maximal number of DFAs accept, breaking ties arbitrarily.

Note that these baselines are entirely unparameterized and operate identically regardless of the underlying DFA. This makes them direct comparisons to using LLMs in in-context-learning[4]. We do not consider BRUTEFORCE$_S$ to be a reasonable comparison due to its computational complexity, and instead consider it an upper bound on performance on this particular task. We choose $n$-GRAM baselines as they are are unambiguously representable by transformers (Svete & Cotterell, 2024), so a transformer model should be able to match their performance.

**Transducer Task**  We have similar baselines for the Transducer task.

- NULL$_T$ baseline: for a given DFA, whichever of the following strategies produces a higher accuracy: always predict 0 or always predict 1.

---

like the one in Figure 1 where there are no "resets" (sequences of symbols that necessarily lead to a particular state), a model must be capable of hollistically evaluating the entire sequence, probably requiring a world model. Many other DFAs contain these resets, but do so in such a way that makes it possible to e.g., recognize that all sequences that end in `a` are in the language, making the problem trivial.

[4]One thing to note is that LLMs are required to determine that they are performing next token prediction on a particular string from a natural language description such as "You are a sequence completion model...," while $n$-GRAM models are programmed to do so. However, we believe all LLMs we evaluate are sophisticated enough to accomplish this without issue.

- $n$-GRAM$_T$ baseline: take the $n-1$ symbols ending at the end of the concatenated transducer sequence (e.g., for $n=5$ and the above example, this would be `1a0a`). If that sequence does not appear elsewhere in the sequence, return the result of the $(n-1)$-GRAM$_T$ baseline. Otherwise, take the token that appears immediately after each occurrence. If there is a majority, return that, otherwise return the last example.

- BRUTEFORCE$_T$: take all possible DFAs with 3 states and 3 symbols. Filter them for ones that match the given transducer sequence. Take this set and predict the next token. Take a majority vote among these, returning 1 by default if there is no majority.

## 4 EXPERIMENTS

We evaluated the open-weight models Llama 3-8B, Llama 3-70B (AI@Meta, 2023), Llama 3.1-8B (AI@Meta, 2024b), Llama 3.1-8B-Instruct (AI@Meta, 2024c), Llama 3.1-70B (AI@Meta, 2024a), Mistral Nemo Minitron 8B (NVIDIA, 2024), Mistral Nemo Base 2407 (Mistral AI, 2024b) and Mistral Nemo Instruct 2407 (Mistral AI, 2024c), Gemma 7B (Google, 2024), Falcon 7B (Almazrouei et al., 2023), Qwen 2.5-7B and Qwen 2.5-32B Team (2024).

We also evaluated the open-weight code models StarCoder2-15B (Lozhkov et al., 2024), Codestral-22B-v0.1 (Mistral AI, 2024a), Deepseek Coder 33B Instruct (Deepseek, 2024), Qwen2.5-Coder-7B, Qwen2.5-Coder-7B-Instruct, and Qwen2.5-Coder-32B-Instruct (Hui et al., 2024).

Finally, we evaluated the proprietary models Claude 3.5 Sonnet (Anthropic, 2024), GPT-3.5-turbo-instruct, GPT-3.5 Chat (turbo-0125) (OpenAI, 2024a), GPT-4o-mini (2024-07-18), GPT 4o (2024-05-13) (OpenAI, 2024b), o3-mini (2025-01-31) (OpenAI, 2025b), and gpt-5 (2025-08-07) (OpenAI, 2025a).

For both tasks, we consider two main prompting formats. BASIC provides no context, presenting the problem as a generic sequence generation or next-token prediction task, where output is provided immediately following the input, with no space to think. BASIC-COT provides the same prompt but asks the model to think step by step and provide an answer. These prompts test ICL, presenting the task in an unstructured manner and requiring the model to learn the problem structure via induction. Our main results are the maximum over these two prompting strategies.

We also provide three "control" prompting formats where information on the problem structure is provided. MORE-EXPL explains that the strings are generated from a simple grammar, but is otherwise identical to BASIC. This remains a sequence generation/next token prediction task. DFA-COT provides the full structure of the latent language, stating that it is a 3-state DFA, and additionally invokes chain-of-thought reasoning to help the model reason over the task. RED-GREEN casts the tasks as independent word problems that describe the underlying grammar structure without relying on world knowledge about DFAs and regular languages. It describes an N-state DFA as a house with N rooms, each of which has 3 portals that deterministically go to other rooms (or back to the same room), where the walls of each room are red or green (mirroring transducer output symbols 0 and 1). Similarly to DFA-COT, the model is given space to show work before providing a tagged answer.

We produce versions of each of these prompts for each task, denoting these with a subscript $_S$ for sequence completion prompts and $_T$ for transducer prompts. Full listings of these prompts can be found in Appendix H. While no finite set of prompts will be fully sufficient to capture all possible model behavior, we believe maximizing over both BASIC prompts allows both models that perform best at next-token-prediction and those that perform better in chain-of-thought reasoning to do their best.

For each open weight model, we used a local VLLM (Kwon et al., 2023) server for evaluation and always evaluated on 1000 distinct DFAs. For GPT-4o and Claude, o3-mini, and gpt-5, we evaluated on 30 DFAs due to computation costs. (Due to greater interest in o3-mini's performance on RED-GREEN$_T$, we used 100 to get a more precise estimate). For gpt-3.5 and gpt-4o-mini, we evaluated on 100 DFAs. All models were evaluated with temperature 0, except reasoning models o3-mini and gpt-5, which do not support a custom temperature.

| Model | Size | IT? | Code? | Sequence Completion | SR | Transducer | TR |
|---|---|---|---|---|---|---|---|
| **Baselines** | | | | | | | |
| BRUTEFORCE | – | | | 100.0 (99.9–100.0) | 1 | 96.4 (96.2–96.7) | 1 |
| 6-GRAM | – | | | **91.7 (91.0–92.4)** | 2 | **93.5 (93.1–93.9)** | 2 |
| 5-GRAM | – | | | 91.2 (90.4–91.9) | 3 | 93.4 (93.0–93.7) | 3 |
| 4-GRAM | – | | | 89.6 (88.7–90.4) | 4 | 91.1 (90.6–91.6) | 4 |
| 3-GRAM | – | | | 87.0 (86.1–87.8) | 5 | 87.0 (86.4–87.6) | 19 |
| 2-GRAM | – | | | 83.3 (82.2–84.2) | 7 | 74.5 (73.6–75.3) | 30 |
| COMMON-SUFFIX | – | | | 84.7 (83.6–85.6) | 6 | – | – |
| RANDOM$_S$/NULL$_T$ | – | | | 53.3 (51.7–54.7) | 32 | 68.9 (68.2–69.6) | 31 |
| **Open Weight Completion** | | | | | | | |
| llama3-8B | 8.0B | | | 73.8 (72.4–75.1) | 22 | 87.5 (86.9–88.0) | 18 |
| llama3-70B | 70.6B | | | 71.4 (70.0–72.7) | 29 | 87.7 (87.2–88.3) | 15 |
| llama3.1-8B-Instruct | 8.0B | ✓ | | 75.3 (74.0–76.6) | 19 | 85.9 (85.3–86.5) | 22 |
| llama3.1-8B | 8.0B | ✓ | | 75.2 (73.8–76.3) | 20 | 88.0 (87.5–88.6) | 10 |
| llama3.1-70B | 70.0B | ✓ | | 71.8 (70.4–73.1) | 28 | 87.7 (87.2–88.2) | 17 |
| qwen-2.5-7B | 7.6B | | | 73.5 (72.1–74.8) | 24 | **88.7 (88.2–89.2)** | 5 |
| qwen-2.5-32B | 32.5B | | | 76.8 (75.5–78.0) | 16 | 88.3 (87.8–88.8) | 7 |
| mistral-nemo-minitron-8B | 8.4B | | | **78.7 (77.5–79.8)** | 13 | 88.6 (88.0–89.1) | 6 |
| mistral-nemo-base-12B | 12.2B | | | 75.5 (74.3–76.6) | 18 | 87.9 (87.4–88.4) | 13 |
| mistral-nemo-instruct-12B | 12.2B | ✓ | | 72.2 (70.9–73.4) | 27 | 88.0 (87.5–88.5) | 11 |
| gemma-7b | 8.5B | | | 72.6 (71.3–73.7) | 25 | 82.1 (81.4–82.7) | 27 |
| falcon-7b | 7.2B | | | 69.0 (67.6–70.2) | 30 | 84.9 (84.3–85.5) | 24 |
| **Open Weight Code** | | | | | | | |
| starcoder2-15b | 16.0B | | ✓ | 73.5 (72.0–74.7) | 23 | 87.7 (85.8–89.5) | 16 |
| codestral-22B | 22.2B | | ✓ | 78.0 (76.8–79.1) | 14 | 86.6 (86.0–87.1) | 21 |
| deepseek-coder-33b-instruct | 33.3B | ✓ | ✓ | 76.7 (75.3–77.8) | 17 | 85.6 (85.0–86.2) | 23 |
| qwen-2.5-coder-7B | 7.6B | | ✓ | **79.5 (78.4–80.5)** | 10 | 88.2 (87.6–88.7) | 9 |
| qwen-2.5-coder-instruct-7B | 7.6B | ✓ | ✓ | 79.5 (78.3–80.5) | 11 | **88.3 (87.8–88.8)** | 8 |
| qwen-2.5-coder-instruct-32B | 32.8B | ✓ | ✓ | 79.2 (78.0–80.3) | 12 | 87.9 (87.4–88.4) | 12 |
| **Proprietary** | | | | | | | |
| gpt-3.5-instruct | ? | ✓ | | 67.3 (63.1–71.5) | 31 | **87.8 (85.9–89.6)** | 14 |
| gpt-3.5-chat | ? | ✓ | | N/A | – | 66.8 (63.4–69.8) | 32 |
| gpt-4o-mini | ? | ✓ | | 72.4 (68.1–76.3) | 26 | 79.8 (77.3–82.2) | 28 |
| gpt-4o | ? | ✓ | | 74.8 (69.3–80.4) | 21 | 83.7 (80.1–86.9) | 25 |
| claude-3.5 | ? | ✓ | | **82.8 (77.5–87.5)** | 8 | 86.9 (83.3–90.0) | 20 |
| o3-mini | ? | ✓ | | 81.1 (76.0–85.8) | 9 | 74.7 (70.7–78.8) | 29 |
| gpt-5 | ? | ✓ | | 77.9 (71.6–84.0) | 15 | 83.6 (79.9–87.1) | 26 |

Table 1: Results for our experiments on both the Transducer and Sequence completion tasks. Each cell contains the mean performance across DFAs for the best-performing BASIC prompt (see Table 2 for details), with 95% confidence intervals of the mean in parentheses. "N/A" means the model returned an invalid result $\geq 25\%$ of the time. (IT = Instruction-Tuned, TR/SR = rank on each task.)

| Model | BASIC | BASIC-COT | MORE-EXPL | DFA-COT | RED-GREEN |
|---|---|---|---|---|---|
| **Sequence Completion** | | | | | |
| gpt-4o-mini | **72.4 (68.1–76.3)** | 60.0 (55.8–64.4) | 70.5 (66.4–74.6) | 58.0 (53.4–62.4) | 59.1 (54.9–63.2) |
| gpt-4o | 72.1 (65.9–78.2) | **74.8 (69.3–80.4)** | N/A | 67.4 (60.8–73.8) | 74.4 (69.9–78.6) |
| claude-3.5 | N/A | 82.8 (77.5–87.5) | N/A | **84.0 (79.3–88.4)** | 80.0 (74.9–85.2) |
| o3-mini | N/A | **81.1 (76.0–85.8)** | N/A | 58.2 (49.6–66.8) | 69.8 (64.4–75.0) |
| gpt-5 | 77.9 (71.6–84.0) | 75.1 (68.6–81.7) | 68.9 (61.0–76.7) | 66.0 (58.9–72.8) | **87.5 (83.1–91.5)** |
| **Transducer** | | | | | |
| gpt-4o-mini | **79.8 (77.3–82.2)** | 66.5 (64.3–68.7) | 76.7 (74.2–79.3) | 65.2 (63.1–67.4) | 74.5 (72.0–77.0) |
| gpt-4o | **83.7 (80.1–86.9)** | 67.8 (62.4–73.2) | 82.6 (79.1–85.9) | 67.8 (63.1–72.3) | 82.6 (78.8–86.3) |
| claude-3.5 | 86.9 (83.3–90.0) | 74.2 (70.0–78.3) | **87.1 (83.9–90.2)** | 76.4 (72.9–79.9) | 82.9 (78.9–86.9) |
| o3-mini | 72.8 (68.4–77.3) | 74.7 (70.7–78.8) | 74.7 (70.3–79.2) | 86.1 (83.9–88.4) | **92.4 (91.3–93.5)** |
| gpt-5 | 83.6 (79.1–87.7) | 83.6 (79.9–87.1) | 85.2 (81.0–88.8) | **96.7 (95.3–98.0)** | 96.6 (95.4–97.8) |

Table 2: Results for models where we investigated multiple prompts (we only used BASIC on other models). We bold the best prompt for each model.

## 5 RESULTS

Main results for all tasks are presented in Table 1. We ignore non-answers, i.e., if for a given DFA a model gets 25 correct answers, 1 incorrect answer, and responds with an unparseable result on 4, this counts as a 25/26, not a 25/29. We then aggregate across DFAs. All comparisons involving 4-GRAM, 5-GRAM, and 6-GRAM to all other models are statistically significant (see Appendix G for details).

### 5.1 SEQUENCE COMPLETION

As seen in Table 1, this task is nearly always fully determined, that is, it can be solved with $\sim100\%$ accuracy in theory, as demonstrated by BRUTEFORCE$_S$ results. Of course, BRUTEFORCE$_S$ is extremely computationally expensive, and, as such, we primarily focus on the $n$-GRAM$_S$ heuristics as our baselines. Still, we find that $n$-GRAM$_S$ heuristics tend to outperform LLMs.

As seen in Table 2, we find that giving the model the opportunity to logically reason about the prompt via chain-of-thought and present a conclusion has inconsistent results. Specifically, we find that for `gpt-4o-mini`, immediately predicting a next token seems to be better, while for `gpt-4o` and `gpt-5` there is no large effect. `claude-3.5` and `o3-mini` are unable to answer the BASIC$_S$ prompt at all, but outperform other proprietary models when using BASIC-COT$_S$. In this task, revealing the problem structure appears to not have a massive effect on performance, with `gpt-5` being the only model to incorporate this information into a statistically significantly improved performance, and even then only in the Red-Green$_S$ word problem prompt (still underperforming 4-GRAM$_S$).

Additionally, we find that in this task, code-specific open-weight models tend to perform better than sequence completion models, suggesting some generalized ability to produce strings from novel languages demonstrated by example. Overall, the relative performances of LLMs and prompts generally comport to heuristics on which models and prompting strategies should work best (with the notable exception of gpt-5). Nonetheless, LLMs underperform simple $n$-GRAM heuristics.

One potential problem with using this task for cross-model comparisons is the relevance of tokenization. We found that forcing uniform tokenization by using commas in the prompt uniformly reduced accuracy, see Appendix E.1 for details; we confirmed that the LLMs we investigate could reason about strings without commas, see Appendix E.2 for details.

### 5.2 TRANSDUCER

Unlike sequence completion, this task is not fully determined, with BRUTEFORCE$_T$ getting 96.4% accuracy. Comparisons are still valid as all models see the same fraction of unsolvable instances.

We find that in general all LLMs underperform a 4-GRAM$_T$ model, demonstrating that they are unable to adequately solve this task. The relative performance of the models also does not correspond to their overall scale, with open weight LLama-3 and Mistral Nemo 8B parameter models outperforming much larger proprietary models. Even within a model class we find no clear pattern: all other GPT models are outperformed by GPT 3.5, Llama 3-70B has similar performance to Llama 3-8B, and the Mistral Nemo 12B models perform similarly to Nemo Minitron 8B. Coding models also demonstrate no advantage on this task.

The generally lower performance of chat-oriented models suggests this task is better suited to non-chat models. More specifically, as seen in Table 2, our BASIC-COT$_T$ prompt results in underperformance by all non-reasoning models, suggesting that models are generally most able to solve this task when it is a simple next-token-prediction task. Providing the problem structure also does not help non-reasoning models improve substantially, but does allow `o3-mini` to perform well, and `gpt-5` to completely solve the task (achieving parity with BRUTEFORCE), demonstrating that reasoning models' underperformance at the original ICL language reasoning task is not due to the underlying difficulty of the task itself.

We conclude that LLM ICL is unable to perform well at language inference. This failure cannot be attributed to a lack of world modeling, as $n$-GRAM$_T$ models do not construct world models. Instead,

it seems the LLMs are unable to detect patterns when those patterns are drawn from an unfamiliar and unknown source, even a relatively simple one.

## 5.3 COMPARISON OF BENCHMARKS

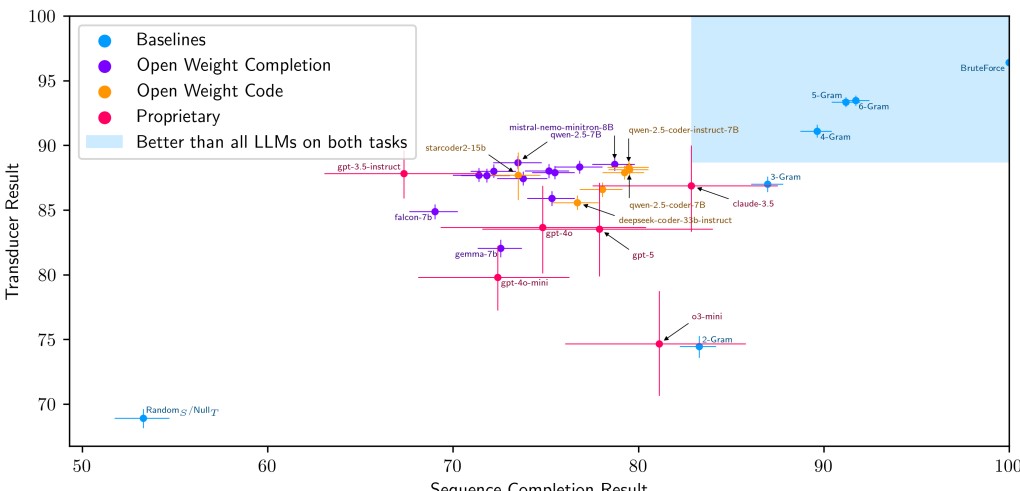

Figure 3: Transducer and sequence completion results plotted against each other. Points are the mean over several DFAs, with 95% confidence intervals. Points are colored by model type, with the best and worst model by each metric in each category labeled, as well as all baseline & proprietary models.

Figure 3 displays the relationship between model performance on the Sequence Completion and Transducer benchmarks. While at a high level, there is a positive correlation between the two, there are a few notable differences. For one, the Code models perform notably better than other open weight models on Sequence Completion, but not on Transducer. Additionally, on Transducer, a ceiling on performance is observed, where LLMs cluster together between $3\text{-}\mathrm{GRAM}_T$ and $4\text{-}\mathrm{GRAM}_T$ performance; this clustering does not appear on the Sequence Completion benchmark.

## 6 CONCLUSION

Our findings highlight significant weaknesses in large language models' ability to in-context-learn entirely novel language reasoning problems, even simple ones solely involving next-token prediction on basic languages recognized by 3-state DFAs. These results, combined with that of previous work demonstrating that large language models can quite accurately perform a variety of language tasks, suggests that LLMs solve language problems via a mechanism distinct from general language reasoning ability. Our use of n-gram baselines and next-token prediction tasks allows us to exclude the possibility that the issue is primarily related to LLMs' lack of world modeling or any inherent limitations of next-token prediction models. We believe that while LLMs have learned individual models of particular languages, our results suggest they have not learned a general theory of language.

Interestingly, in our transducer experiments, directly predicting the next token rather than explicitly reasoning through the problem works better, except for reasoning models where they perform similarly. While our conclusions are limited by the finite nature of our prompt set, this suggests that they do, in fact, possess some latent understanding of language, but this understanding is inferior to basic n-gram models for $n > 3$.

One potential goal for foundation models is to replace all machine learning with ICL. Our results suggest that current models are not progressing towards this goal.

IMPACT STATEMENT

Aside from the social consequences of this work as related to advancing the field of Machine Learning in general, this work has the goal of advancing the field of benchmarks in Machine Learning. While we view this as a positive objective, as it ensures that models are being evaluated fairly, it might have negative consequences insofar as benchmarking techniques might be best left unpublished to prevent deliberate or unintentional overfitting.

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

## A  DETAILS ON SAMPLING

### A.1  SAMPLING OF DFAS

We use rejection sampling to sample DFAs. Specifically, we uniformly sample a start state, then for each (source state, symbol) pair, we sample a post-transition state. We also randomly assign each state to be accept or reject with probability 50%. We then reject any DFA that has all accept or all reject states (so only DFAs with 1 or 2 accept states are allowed), or for which certain states are unreachable from the start state.

### A.2  SAMPLING OF SEQUENCE COMPLETION TASKS

To sample a sequence completion task, we first sample a DFA as described in Appendix A.1.

To sample a task instance, we sample example sequences and distinct prefix. Each example sequence is sampled uniformly from the space of $\{a, b, c\}^{10}$ and then rejected if the DFA does not accept the sequence. Our distinct prefix and completion are sampled uniformly from $\{a, b, c\}^5 \times \{a, b, c\}^5$, and are rejected if the DFA does not accept the concatenation of the two, or if the prefix is the prefix of any of the previous sequences. We then discard the completion. If we, at any point, reject 50 sequences when attempting to sample a sequence or prefix, we return an error.

We run a "pilot" sampling for a DFA to ensure that it is valid, in which we sample an instance as described above. If there is an error in sampling this pilot instance, we reject the DFA. Otherwise, we proceed to sample our task instances. At this stage, if there is an error in sampling, we reject the instance rather than the DFA. This pilot sample rejection procedure leads to a slight bias towards 2-accept state DFAs over 1-accept state DFAs, as measured by the RANDOM$_S$ baseline.

### A.3  SAMPLING OF TRANSDUCER TASKS

We sample a DFA as described in Appendix A.1, and then sample random sequences (30 in our experiments) and generate transducer traces. If every transducer trace ends with a 0 or every trace ends with a 1, we reject the DFA and resample.

# B    TRANSDUCER RESULTS BY DIFFICULTY CLASS

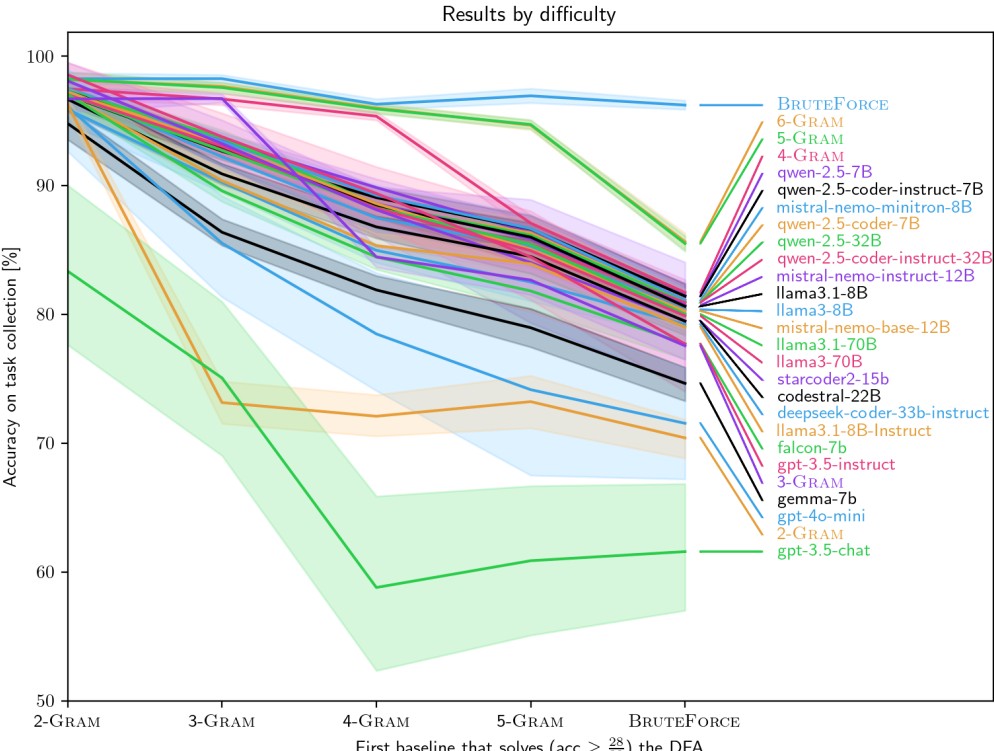

Figure 4: Transducer results by difficulty class. We classify each DFA based on which of the baselines first achieves a score of 28/30 on the given instances. 6-GRAM is excluded as it has very similar performance to 5-GRAM. Each model's best prompt results are plotted, with 95% confidence intervals, for all models with at least 100 DFAs; those with 10 or 30 had error bars too large to make this analysis useful.

Figure 4 displays results by difficulty level, as judged by the smallest $n$-GRAM model that can solve a particular task. All models behave roughly monotonically, performing more poorly as difficulty increases. Additionally, we find that the best models continue to perform similarly to 4-GRAM for tasks that 4-GRAM does not perfectly solve.

## C  VARYING NUMBER OF EXAMPLES AND NUMBER OF STATES

### C.1  VARYING NUMBER OF EXAMPLES

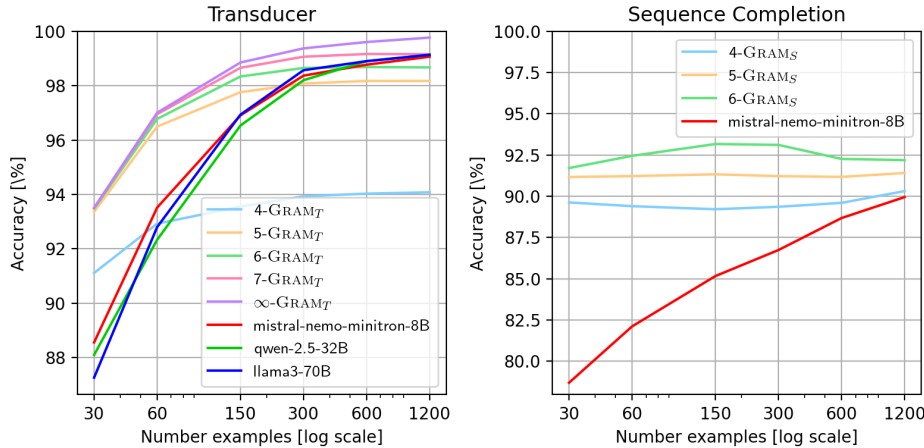

Figure 5: Accuracy by number of examples, as we vary the number from 30 to 1200 (note the log scale). ∞-GRAM is a model that finds the longest sequence that matches the ending of the sequence and copies the following token.

We varied the number of examples parameter from 30 to 1200, investigating specifically `mistral-nemo-minitron-8B` as it is a better performing model on the transducer task, as well as `qwen-2.5-32B` and `llama3-70B` to give an example of a larger model (only on Transducer, as the memory consumption was too great to increase the number of samples on Sequence Completion). We find that for the Sequence Completion task, the $n$-GRAM models do not generally improve as the number of examples increases, while the LLM does; however, the LLM remains below 4-GRAM performance at all points. In the Transducer experiment, both the n-GRAM models and the LLM improve, with the LLM crossing 4-GRAM and 5-Gram performance and eventually roughly matching 6-GRAM performance. However, there is now an increased gap between 6-GRAM and 7-GRAM that did not exist at 30 (we did not include 7-GRAM or above in the main table for this reason). Overall, it is possible that the larger number of instances of the "correct n-GRAM" appearing (i.e., the suffix of the prompt followed by the correct answer) causes the model to be better at producing predictions. As the LLMs at no point outperform the best n-GRAM and the ∞-gram gets saturating accuracy, we do not believe that this alters our overall conclusions. `qwen-2.5-32B` and `llama3-70B` catch up with `mistral-nemo-minitron-8B` for larger numbers of examples, but in general their performances are quite similar throughout, suggesting the effect seen here is largely invariant to model scale.

## C.2 VARYING NUMBER OF STATES

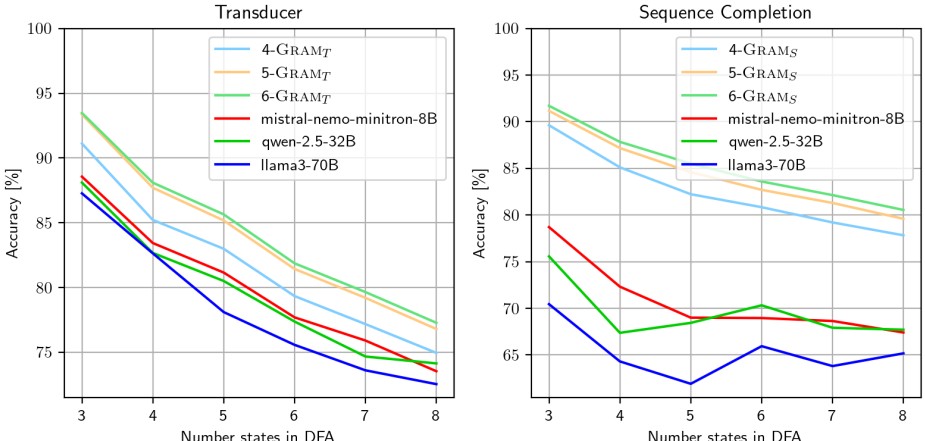

Figure 6: Accuracy by number of states.

As seen in Figure 6, as the number of states increases, all models' performance reduces, to a similar degree. This indicates that more states makes the problem overall more difficult, without particularly benefiting either the n-Gram models or LLMs.

# D   NONZERO TEMPERATURE

| Model | Prompt | Zero Temp | Nonzero Temp | Difference |
|---|---|---|---|---|
| **Sequence Completion** | | | | |
| mistral-nemo-minitron-8B | BASIC | 78.70% (77.49% – 79.79%) | 77.67% (76.51% – 78.76%) | -1.04% (-1.49% – -0.63%) |
| claude-3.5 | COT | 84.00% (79.33% – 88.44%) | 84.22% (79.56% – 89.00%) | 0.22% (-2.11% – 2.33%) |
| claude-3.5 | RED-GREEN | 80.00% (74.89% – 85.22%) | 80.78% (75.00% – 86.11%) | 0.78% (-2.11% – 3.44%) |
| **Transducer** | | | | |
| mistral-nemo-minitron-8B | BASIC | 88.56% (88.05% – 89.08%) | 88.17% (87.64% – 88.68%) | -0.39% (-0.58% – -0.22%) |
| claude-3.5 | BASIC | 86.89% (83.33% – 90.00%) | 87.00% (83.33% – 90.11%) | 0.11% (-0.67% – 0.89%) |
| claude-3.5 | MORE-EXPL | 87.11% (83.88% – 90.22%) | 86.89% (83.22% – 90.11%) | -0.22% (-1.11% – 0.67%) |
| claude-3.5 | COT | 76.44% (72.89% – 79.89%) | 78.11% (74.66% – 81.33%) | 1.67% (-0.56% – 3.78%) |
| claude-3.5 | RED-GREEN | 82.89% (78.89% – 86.89%) | 82.78% (78.89% – 86.33%) | -0.11% (-2.00% – 1.78%) |

Table 3: Results varying temperature. Second column is a temperature of 0.1, third column is differences. In all columns, we annotate a 95% confidence interval, using paired differences for the third column.

In order to determine whether a small nonzero temperature might lead to better results, we investigated using a temperature of 0.1 for mistral-nemo-minitron-8B and claude-3.5 (these two chosen for the reasons described in Section I, notably o3-mini/gpt-5 already are using nonzero temperatures). We find that a temperature of 0.1 does not significantly change performance, resulting in no significant change for any model/task/prompt combination. The largest improvement we observe is claude-3.5 on COT on the Transducer task, which gains 1.7%, from 76.4% to 78.1% (still not enough to make it the best prompt for claude-3.5).

# E  TOKENIZATION

| Model | BASIC$_S$ | BASIC-COMMAS$_S$ |
|---|---|---|
| **qwen-2.5-coder-7B** | **79.5 (78.4–80.5)** | **60.7 (59.3–62.1)** |
| qwen-2.5-coder-instruct-7B | 79.5 (78.3–80.5) | 55.5 (54.0–56.9) |
| qwen-2.5-coder-instruct-32B | 79.2 (78.0–80.3) | 55.2 (53.7–56.7) |
| mistral-nemo-minitron-8B | 78.7 (77.5–79.8) | 59.3 (57.9–60.8) |
| codestral-22B | 78.0 (76.8–79.1) | 59.0 (57.5–60.3) |
| qwen-2.5-32B | 76.8 (75.5–78.0) | 60.3 (58.9–61.6) |
| deepseek-coder-33b-instruct | 76.7 (75.3–77.8) | 54.9 (53.0–56.8) |
| mistral-nemo-base-12B | 75.5 (74.3–76.6) | 60.6 (59.1–62.2) |
| llama3.1-8B-Instruct | 75.3 (74.0–76.6) | 56.3 (54.4–58.1) |
| llama3.1-8B | 75.2 (73.8–76.3) | 61.1 (59.8–62.5) |
| llama3-8B | 73.8 (72.4–75.1) | 61.5 (60.2–62.9) |
| starcoder2-15b | 73.5 (72.0–74.7) | 58.2 (56.7–59.8) |
| qwen-2.5-7B | 73.5 (72.1–74.8) | 57.0 (55.5–58.5) |
| gemma-7b | 72.6 (71.3–73.7) | 54.0 (51.9–56.0) |
| gpt-4o-mini | 72.4 (68.1–76.3) | 64.1 (59.5–68.3) |
| mistral-nemo-instruct-12B | 72.2 (70.9–73.4) | 58.2 (56.4–59.8) |
| gpt-4o | 72.1 (65.9–78.2) | 66.8 (58.5–74.8) |
| llama3.1-70B | 71.8 (70.4–73.1) | 57.7 (56.1–59.2) |
| llama3-70B | 71.4 (70.0–72.7) | 56.4 (54.7–58.0) |
| falcon-7b | 69.0 (67.6–70.2) | 56.1 (54.5–57.6) |
| gpt-3.5-instruct | 67.3 (63.1–71.5) | 52.3 (46.5–57.9) |
| o3-mini | N/A | N/A |
| claude-3.5 | N/A | N/A |
| gpt-3.5-chat | N/A | N/A |

Table 4: Results on Sequence Completion Task. We compare BASIC$_S$ to the comma-variant BASIC-COMMAS$_S$.

## E.1  SEQUENCE COMPLETION TASK PROMPT WITH COMMAS

To avoid tokenization differences with models, we also investigate a version of our Sequence Completion prompt that uses spaces and commas between the elements of the sequence. Unfortunately, results using this prompt were uniformly worse than results on the prompt without spaces and commas. Table 4 shows the results on a variety of models. All are worse with commas than without.

## E.2  DIRECTLY CONFIRMING MODELS CAN READ SEQUENCES OF LETTERS

To fully exclude the possibility that models are unable to read sequences of letters, we perform the following experiment: we take the regular expression `ab(abc)+` and directly provide it to the model, then ask the model to test it on a string (one string provided per query), using the prompt `I will give you a string. Tell me whether it matches the following regular expression: 'âb(abc)+$' (without quotes). Just answer YES or NO.` on one line, followed by a string on the next. We sample 100 random strings of length 2-17, via the following procedure (1) sample a random valid string uniformly (2) with 50% probability, randomly mutate one of the elements of the string to a different character. All strings are thus either correct or near-correct. We also add `Answer (YES or NO):` on a third line for non-chat models to encourage them to provide a response rather than another string.

Results for this are presented in Table 5. While not all models perform well at this task, performances follow roughly what one might expect from standard benchmarks (rather than the Sequence Completion results in the paper). Frontier proprietary models perform extremely well, larger open weight models tend to perform well, and smaller open weight models are more hit-and-miss (some performing well, some poorly).

| Model | Accuracy | Non-response |
|---|---|---|
| llama3-8B | 66% | |
| llama3-70B | 80% | 26% |
| llama3.1-8B-Instruct | 83% | |
| llama3.1-8B | 89% | |
| llama3.1-70B | 86% | 36% |
| qwen-2.5-7B | 62% | |
| qwen-2.5-32B | 100% | |
| mistral-nemo-instruct-12B | 99% | |
| gemma-7b | 67% | 15% |
| starcoder2-15b | 62% | |
| codestral-22B | 95% | |
| qwen-2.5-coder-7B | 100% | |
| qwen-2.5-coder-instruct-7B | 56% | |
| qwen-2.5-coder-instruct-32B | 96% | |
| gpt-3.5-instruct | 67% | |
| gpt-3.5-chat | 82% | |
| gpt-4o-mini | 100% | |
| gpt-4o | 100% | |
| claude-3.5 | 100% | |
| o3-mini | 100% | |
| gpt-5 | 100% | |

Table 5: Results on Regex task. As in the rest of this paper, accuracies are computed ignoring non-response. Models mistral-nemo-minitron-8B, mistral-nemo-base-12B, falcon-7b, deepseek-coder-33b-instruct have non-response rates over 98%.

The proprietary models newer than the GPT-3 series get 100% accuracy on this task, while performance is lower for the open weight models. There is no obvious relationship between models that perform well at this particular task and models that perform well at Sequence Completion, indicating that to whatever degree models are performing poorly at this test task, it is not because they are unable to process the string. Notably, o3-mini, gpt-4o, and gpt-4o-mini all perform fairly poorly at the Sequence Completion task, placing below median, but all achieve 100% on this task.

Open weight models are more mixed, with larger ones tending to do nearly as well as newer proprietary models (notable exception being llama3-70B), but smaller ones occasionally performing well and occasionally performing poorly.

# F  MODEL NON-ANSWERS

Table 6 depicts the percentage of model non-answers by model and prompt. In general, this distribution is highly bimodal, with values always being either below 9% or above 97%.

The only prompt-vs-prompt orderings that are changed by scoring non-answers as 0 are that, on Sequence Completion, BASIC$_S$ rises above RED-GREEN$_S$ for gpt-4o, making it the best prompt; and that on Transducer, RED-GREEN$_T$ for gpt-4o-mini rises above MORE-EXPL$_T$ (though still behind BASIC$_T$. The qualitative conclusions about next token prediction vs chain of thought results remain the same.

The only change to relative model ordering is that on Sequence Completion, gpt-4o drops 8 ranks, from 17th place to 25nd place, being passed by several open weight models, gpt-4o-mini, and o3-mini. No change occurs on the transducer results. Qualiative conclusions about model ordering remain the same.

| Model | BASIC | BASIC-COT | MORE-EXPL | DFA-COT | RED-GREEN |
|---|---|---|---|---|---|
| **Sequence Completion** | | | | | |
| llama3-8B | 0.0 (0.0–0.0) | – | – | – | – |
| llama3-70B | 0.0 (0.0–0.0) | – | – | – | – |
| llama3.1-8B-Instruct | 0.0 (0.0–0.0) | – | – | – | – |
| llama3.1-8B | 0.0 (0.0–0.0) | – | – | – | – |
| llama3.1-70B | 0.0 (0.0–0.0) | – | – | – | – |
| mistral-nemo-minitron-8B | 0.0 (0.0–0.0) | – | – | – | – |
| mistral-nemo-base-12B | 0.0 (0.0–0.0) | – | – | – | – |
| mistral-nemo-instruct-12B | 0.0 (0.0–0.0) | – | – | – | – |
| gemma-7b | 0.0 (0.0–0.0) | – | – | – | – |
| falcon-7b | 0.0 (0.0–0.0) | – | – | – | – |
| starcoder2-15b | 0.0 (0.0–0.0) | – | – | – | – |
| codestral-22B | 0.0 (0.0–0.0) | – | – | – | – |
| deepseek-coder-33b-instruct | 0.0 (0.0–0.0) | – | – | – | – |
| qwen-2.5-coder-instruct-7B | 0.0 (0.0–0.0) | – | – | – | – |
| qwen-2.5-7B | 0.0 (0.0–0.0) | – | – | – | – |
| qwen-2.5-32B | 0.0 (0.0–0.0) | – | – | – | – |
| qwen-2.5-coder-instruct-32B | 0.0 (0.0–0.0) | – | – | – | – |
| qwen-2.5-coder-7B | 0.0 (0.0–0.0) | – | – | – | – |
| gpt-3.5-instruct | 2.5 (1.9–3.0) | – | – | – | – |
| gpt-3.5-chat | 99.9 (99.7–100.0) | – | – | – | – |
| gpt-4o-mini | 0.0 (0.0–0.0) | 0.3 (0.2–0.5) | 0.0 (0.0–0.0) | 1.0 (0.6–1.4) | 0.2 (0.1–0.4) |
| gpt-4o | 4.4 (3.3–5.7) | 1.9 (1.1–2.8) | 100.0 (100.0–100.0) | 5.0 (3.8–6.2) | 8.4 (6.6–10.2) |
| claude-3.5 | 99.7 (99.2–100.0) | 0.1 (0.0–0.3) | 97.8 (96.9–98.6) | 0.0 (0.0–0.0) | 0.0 (0.0–0.0) |
| o3-mini | 80.2 (78.0–82.3) | 3.8 (2.6–5.2) | 91.6 (89.9–93.2) | 5.7 (4.1–7.2) | 0.4 (0.0–1.0) |
| gpt-5 | 0.0 (0.0–0.0) | 0.0 (0.0–0.0) | 8.2 (4.7–12.1) | 5.2 (3.1–7.4) | 0.4 (0.1–0.9) |
| **Transducer** | | | | | |
| llama3-8B | 0.0 (0.0–0.0) | – | – | – | – |
| llama3-70B | 0.0 (0.0–0.0) | – | – | – | – |
| llama3.1-8B-Instruct | 0.0 (0.0–0.0) | – | – | – | – |
| llama3.1-70B | 0.0 (0.0–0.0) | – | – | – | – |
| llama3.1-8B | 0.0 (0.0–0.0) | – | – | – | – |
| starcoder2-15b | 0.0 (0.0–0.0) | – | – | – | – |
| codestral-22B | 0.0 (0.0–0.0) | – | – | – | – |
| deepseek-coder-33b-instruct | 0.0 (0.0–0.0) | – | – | – | – |
| qwen-2.5-coder-7B | 0.0 (0.0–0.0) | – | – | – | – |
| qwen-2.5-coder-instruct-7B | 0.0 (0.0–0.0) | – | – | – | – |
| qwen-2.5-7B | 0.0 (0.0–0.0) | – | – | – | – |
| qwen-2.5-32B | 0.0 (0.0–0.0) | – | – | – | – |
| qwen-2.5-coder-instruct-32B | 0.0 (0.0–0.0) | – | – | – | – |
| mistral-nemo-minitron-8B | 0.0 (0.0–0.0) | – | – | – | – |
| mistral-nemo-base-12B | 0.0 (0.0–0.0) | – | – | – | – |
| mistral-nemo-instruct-12B | 0.0 (0.0–0.0) | – | – | – | – |
| gemma-7b | 0.0 (0.0–0.0) | – | – | – | – |
| falcon-7b | 0.0 (0.0–0.0) | – | – | – | – |
| gpt-3.5-instruct | 0.0 (0.0–0.1) | – | – | – | – |
| gpt-3.5-chat | 0.1 (0.0–0.3) | – | – | – | – |
| gpt-4o-mini | 1.8 (1.3–2.3) | 0.0 (0.0–0.0) | 5.8 (4.8–6.9) | 0.0 (0.0–0.0) | 0.7 (0.4–1.0) |
| gpt-4o | 0.0 (0.0–0.0) | 0.0 (0.0–0.0) | 0.0 (0.0–0.0) | 0.0 (0.0–0.0) | 0.0 (0.0–0.0) |
| claude-3.5 | 0.0 (0.0–0.0) | 0.0 (0.0–0.0) | 0.0 (0.0–0.0) | 0.0 (0.0–0.0) | 0.0 (0.0–0.0) |
| o3-mini | 0.1 (0.0–0.3) | 0.0 (0.0–0.0) | 0.4 (0.1–0.9) | 0.0 (0.0–0.0) | 0.0 (0.0–0.1) |
| gpt-5 | 0.0 (0.0–0.0) | 0.0 (0.0–0.0) | 0.0 (0.0–0.0) | 0.0 (0.0–0.0) | 0.7 (0.1–1.4) |

Table 6: Model non-answers, as a percentage of all prompt responses. A non-response is not included in accuracy computations for Table 1 or Table 2, but whenever it rises above 25%, N/A is placed in those tables.

## G SIGNIFICANCE

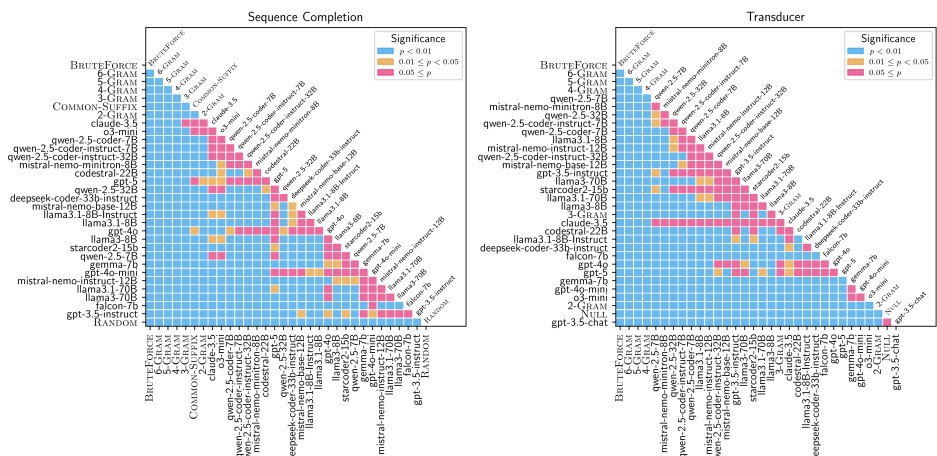

Figure 7: Significance of comparisons between rows of Table 1. Results in blue and orange are significant, results in pink are not.

Figure 7 shows which comparisons between rows of Table 1 are significant. Significance computations are performed by running a 2-tailed bootstrap significance test on paired (by DFAs) differences.

# H  PROMPT LISTINGS

## H.1  SUMMARIES

Table 7 contains summaries of each prompt.

## H.2  FULL EXAMPLE LISTINGS

| Prompt | $T$ | $S$ |
|---|---|---|
| BASIC | You are a sequence completion model. Output the next element of the sequence, and nothing else.

\<TRANSDUCER PREFIX\>, | The following strings come from an alien language that follows a simple grammar. Infer the alien grammar using the example strings. Then, add a suffix to the final string using between 1 and 5 characters such that the full string follows the grammar. Output only the necessary suffix to complete the final string, and nothing else.

\<EXAMPLES\>
\<PREFIX\> |
| BASIC-COT | You are a sequence completion model. Reason step by step, and then output the next output integer using \<answer\> tags, like \<answer\>0\</answer\>.

Input sequence: \<TRANSDUCER PREFIX\>
Output sequence: | The following strings come from an alien language that follows a simple grammar. Infer the alien grammar using the example strings. Then, add a suffix to the final string using between 1 and 5 characters such that the full string follows the grammar. Reason step by step, and then output the next necessary suffix for this final string, \<answer\> tags, like \<answer\>ab\</answer\>.

\<EXAMPLES\>
\<PREFIX\> |
| MORE-EXPL | You are a sequence completion model. The following sequence is generated from an unknown but consistent grammar. Identify the patterns within the sequence to determine its next element. Output the next element of the sequence, and nothing else.

\<TRANSDUCER PREFIX\>, | I have a 3-state DFA model that outputs either 0 or 1 after each element I input. 1 indicates that the input string thus far results in a "valid" state, and 0 indicates that it does not. I collect a set of valid strings using this DFA, listed below. Infer the underlying DFA model using these strings and complete the final string, using up to n characters, such that it is also a valid string. Output only the necessary suffix to complete the final string, and nothing else.
\<EXAMPLES\>
\<PREFIX\> |
| DFA-COT | A DFA is a finite-state machine that accepts or rejects a given string of symbols, by running through a n-state sequence uniquely determined by the string.

I have a 3-state DFA model that outputs either 0 or 1 after each element I input. 1 indicates that the input string thus far results in a "valid" state, and 0 indicates that it does not. I collect the inputs and outputs into an input sequence and an output sequence. Infer the underlying DFA model to predict the next integer in the output sequence. Reason step by step, and then output the next output integer using \<answer\> tags, like \<answer\>0\</answer\>.

Input sequence: \<TRANSDUCER PREFIX\>
Output sequence: | I have a 3-state DFA model that outputs either 0 or 1 after each element I input. 1 indicates that the input string thus far results in a "valid" state, and 0 indicates that it does not. I collect a set of valid strings using this DFA, listed below. Infer the underlying DFA model using these strings and complete the final string, using up to n characters, such that it is also a valid string. Reason step by step, and then output the next necessary suffix for this final string, \<answer\> tags, like \<answer\>ab\</answer\>.

Given these valid strings:
\<EXAMPLES\>

Complete the following string:
\<PREFIX\> |
| RED-GREEN | ```
You are in a house of rooms and portals. There are 3 rooms in the house, and each room has 3 unique portals labeled A, B, and C. Each portal teleports you to one room of the house (and sometimes the destination is the room the portal is in). Every portal in a given room always behaves the same way.

In this house, each of the rooms look exactly the same, except some of the rooms have red walls and some have green walls. However, there are *three* rooms in total, so you cannot determine which room you are in by color alone, and two rooms of the same color may have portals that behave differently. As you move through the house, at each time step you write down what portal you take and the color of the room you arrive (or stay) in. Based on your notes, predict what color room you will end up in after the last step.

Tag your final answer like \<answer\>color\</answer\>.

You walk through a portal labeled "\<TRANSDUCER PREFIX\>" and end up in a red room.
``` | You are outside a house of rooms and portals. There are 3 rooms in the house, and each room has 3 unique portals labeled a, b, and c. Each portal teleports you to one room of the house (and sometimes the destination is the room the portal is in). Every portal in a given room always behaves the same way.

In this house, each of the rooms look exactly the same, except some of the rooms have red walls and some have green walls. However, there are *3* rooms in total, so you cannot determine which room you are in by color alone, and two rooms of the same color may have portals that behave differently. You've been into this house many times before. Each time, as you move through the house, you write down what series of portals you take and the color of the room you end up in. You have a collection of paths you've taken where you've ended up in a room with green walls, listed below. Given the final incomplete path at the bottom, write a series of up to 5 remaining steps that will cause you to end up in a room with green walls again.

Tag your final answer like \<answer\>ab\</answer\>.

Given these paths that end in a room with green walls:
\<EXAMPLES\>

Complete the following path:
\<PREFIX\> |

### H.2.1 BASIC$_T$

You are a sequence completion model. Output the next element of the sequence, and nothing else.

a, 1, b, 1, a, 1, b, 1, b, 1, c, 0, a, 1, c, 1, a, 1, a, 1, a, 1, c, 1, b, 1, c, 0, c, 1, a, 1, b, 1, b, 1, b, 1, b, 1, a, 1, b, 1, a, 1, a, 1, b, 1, c, 0, a, 1, c, 1, a, 1, b,

### H.2.2 BASIC-COT$_T$

You are a sequence completion model. Reason step by step, and then output the next output integer using <answer> tags, like <answer>0</answer>.

Input sequence: a, b, a, b, b, c, a, c, a, a, a, c, b, c, c, a, b, b, b, b, a, b, a, a, b, c, a, c, a, b
Output sequence: 1, 1, 1, 1, 1, 0, 1, 1, 1, 1, 1, 1, 1, 0, 1, 1, 1, 1, 1, 1, 1, 1, 1, 1, 1, 0, 1, 1, 1,

### H.2.3 MORE-EXPL$_T$

You are a sequence completion model. The following sequence is generated from an unknown but consistent grammar. Identify the patterns within the sequence to determine its next element. Output the next element of the sequence, and nothing else.

a, 1, b, 1, a, 1, b, 1, b, 1, c, 0, a, 1, c, 1, a, 1, a, 1, a, 1, c, 1, b, 1, c, 0, c, 1, a, 1, b, 1, b, 1, b, 1, b, 1, a, 1, b, 1, a, 1, a, 1, b, 1, c, 0, a, 1, c, 1, a, 1, b,

### H.2.4 DFA-COT$_T$

A DFA is a finite-state machine that accepts or rejects a given string of symbols, by running through a n-state sequence uniquely determined by the string.

I have a 3-state DFA model that outputs either 0 or 1 after each element I input. 1 indicates that the input string thus far results in a "valid" state, and 0 indicates that it does not. I collect the inputs and outputs into an input sequence and an output sequence. Infer the underlying DFA model to predict the next integer in the output sequence. Reason step by step, and then output the next output integer using <answer> tags, like <answer>0</answer>.

Input sequence: a, b, a, b, b, c, a, c, a, a, a, c, b, c, c, a, b, b, b, b, a, b, a, a, b, c, a, c, a, b
Output sequence: 1, 1, 1, 1, 1, 0, 1, 1, 1, 1, 1, 1, 1, 0, 1, 1, 1, 1, 1, 1, 1, 1, 1, 1, 1, 0, 1, 1, 1,

### H.2.5 RED-GREEN$_T$

```
You are in a house of rooms and portals. There are 3 rooms in the house, and each room has 3 unique portals labeled A, B, and C. Each portal teleports you to one room of the house (and sometimes the destination is the room the portal is in). Every portal in a given room always behaves the same way.

In this house, each of the rooms look exactly the same, except some of the rooms have red walls and some have green walls. However, there are *three* rooms in total, so you cannot determine which room you are in by color alone, and two rooms of the same color may have portals that behave differently. As you move through the house, at each time step you write down what portal you take and the color of the room you arrive (or stay) in. Based on your notes, predict what color room you will end up in after the last step.

Tag your final answer like <answer>color</answer>.

You walk through a portal labeled "A" and end up in a green room.
Then, you walk through a portal labeled "B" and end up in a green room.
Then, you walk through a portal labeled "A" and end up in a green room.
Then, you walk through a portal labeled "B" and end up in a green room.
Then, you walk through a portal labeled "B" and end up in a green room.
Then, you walk through a portal labeled "C" and end up in a red room.
Then, you walk through a portal labeled "A" and end up in a green room.
Then, you walk through a portal labeled "C" and end up in a green room.
Then, you walk through a portal labeled "A" and end up in a green room.
Then, you walk through a portal labeled "A" and end up in a green room.
Then, you walk through a portal labeled "A" and end up in a green room.
Then, you walk through a portal labeled "C" and end up in a green room.
Then, you walk through a portal labeled "B" and end up in a green room.
Then, you walk through a portal labeled "C" and end up in a red room.
Then, you walk through a portal labeled "C" and end up in a green room.
Then, you walk through a portal labeled "A" and end up in a green room.
Then, you walk through a portal labeled "B" and end up in a green room.
Then, you walk through a portal labeled "B" and end up in a green room.
Then, you walk through a portal labeled "B" and end up in a green room.
Then, you walk through a portal labeled "B" and end up in a green room.
Then, you walk through a portal labeled "A" and end up in a green room.
Then, you walk through a portal labeled "B" and end up in a green room.
Then, you walk through a portal labeled "A" and end up in a green room.
Then, you walk through a portal labeled "A" and end up in a green room.
Then, you walk through a portal labeled "B" and end up in a green room.
Then, you walk through a portal labeled "C" and end up in a red room.
Then, you walk through a portal labeled "A" and end up in a green room.
Then, you walk through a portal labeled "C" and end up in a green room.
Then, you walk through a portal labeled "A" and end up in a green room.
Then, you walk through a portal labeled "B" and end up in a ...
```

### H.2.6 BASIC$_S$

The following strings come from an alien language that follows a simple grammar. Infer the alien grammar using
the example strings. Then, add a suffix to the final string using between 1 and 5 characters such that
the full string follows the grammar. Output only the necessary suffix to complete the final string, and
nothing else.

```
cbcbabbcca
abcaaacbaa
aabccbabbb
bbbccbbbca
aababaccba
aaaacbacac
baacbccbaa
cbbaacabcc
baabaacaab
bbbbbcacab
acaabcbbba
acaacbccac
cacbabcbba
abcbcbcbcc
ccaccccaba
bcbcabbcca
baabacabca
caababacac
bacacaccaa
bcacbbbbca
bcbbbcaccc
ccabbcccbb
bccbcabbca
baacbabcbc
ccacabccab
caacbcaaab
cacbaaccac
aaccbcaabb
abacabcaab
bacbcbcaca
caacb
```

### H.2.7 BASIC-COT$_S$

The following strings come from an alien language that follows a simple grammar. Infer the alien grammar using
the example strings. Then, add a suffix to the final string using between 1 and 5 characters such that
the full string follows the grammar. Reason step by step, and then output the next necessary suffix for
this final string, <answer> tags, like <answer>ab</answer>.

```
cbcbabbcca
abcaaacbaa
aabccbabbb
bbbccbbbca
aababaccba
aaaacbacac
baacbccbaa
cbbaacabcc
baabaacaab
bbbbbcacab
acaabcbbba
acaacbccac
cacbabcbba
abcbcbcbcc
ccaccccaba
bcbcabbcca
baabacabca
caababacac
bacacaccaa
bcacbbbbca
bcbbbcaccc
ccabbcccbb
bccbcabbca
baacbabcbc
ccacabccab
caacbcaaab
cacbaaccac
aaccbcaabb
abacabcaab
bacbcbcaca
caacb
```

### H.2.8 BASIC-COMMAS$_S$

The following strings come from an alien language that follows a simple grammar. Infer the alien grammar using
the example strings. Then, add a suffix to the final string using between 1 and 5 characters such that
the full string follows the grammar. Output only the necessary suffix to complete the final string, and
nothing else.

```
c, b, c, b, a, b, b, c, c, a
a, b, c, a, a, a, c, b, a, a
```

```
a, a, b, c, c, b, a, b, b, b
b, b, b, c, c, b, b, b, c, a
a, a, b, a, b, a, c, c, b, a
a, a, a, a, c, b, a, c, a, c
b, a, a, c, b, c, c, b, a, a
c, b, b, a, a, c, a, b, c, c
b, a, a, b, a, a, c, a, a, b
b, b, b, b, b, c, a, c, a, b
a, c, a, a, b, c, b, b, b, a
a, c, a, a, c, b, c, c, a, c
c, a, c, b, a, b, c, b, b, a
a, b, c, b, c, b, c, b, c, c
c, c, a, c, c, c, c, a, b, a
b, c, b, c, a, b, b, c, c, a
b, a, a, b, a, c, a, b, c, a
c, a, a, b, a, b, a, c, a, c
b, a, c, a, c, a, c, c, a, a
b, c, a, c, b, b, b, b, c, a
b, c, b, b, b, c, a, c, c, c
c, c, a, b, b, c, c, c, b, b
b, c, c, b, c, a, b, b, c, a
b, a, a, c, b, a, b, c, b, c
c, c, a, c, a, b, c, c, a, b
c, a, a, c, b, c, a, a, a, b
c, a, c, b, a, a, c, c, a, c
a, a, c, c, b, c, a, a, b, b
a, b, a, c, a, b, c, a, a, b
b, a, c, b, c, b, c, a, c, a
c, a, a, c, b,
```

### H.2.9   MORE-EXPL$_S$

I have a 3-state DFA model that outputs either 0 or 1 after each element I input. 1 indicates that the input
    string thus far results in a "valid" state, and 0 indicates that it does not. I collect a set of valid
    strings using this DFA, listed below. Infer the underlying DFA model using these strings and complete
    the final string, using up to n characters, such that it is also a valid string. Output only the
    necessary suffix to complete the final string, and nothing else.
cbcbabbcca
abcaaacbaa
aabccbabbb
bbbccbbbca
aababaccba
aaaacbacac
baacbccbaa
cbbaacabcc
baabaacaab
bbbbbcacab
acaabcbbba
acaacbccac
cacbabcbba
abcbcbcbcc
ccaccccaba
bcbcabbcca
baabacabca
caababacac
bacacaccaa
bcacbbbbca
bcbbbcaccc
ccabbcccbb
bccbcabbca
baacbabcbc
ccacabccab
caacbcaaab
cacbaaccac
aaccbcaabb
abacabcaab
bacbcbcaca
caacb

### H.2.10   DFA-COT$_S$

I have a 3-state DFA model that outputs either 0 or 1 after each element I input. 1 indicates that the input
    string thus far results in a "valid" state, and 0 indicates that it does not. I collect a set of valid
    strings using this DFA, listed below. Infer the underlying DFA model using these strings and complete
    the final string, using up to n characters, such that it is also a valid string. Reason step by step,
    and then output the next necessary suffix for this final string, <answer> tags, like <answer>ab</answer
    >.

Given these valid strings:
cbcbabbcca
abcaaacbaa
aabccbabbb
bbbccbbbca
aababaccba
aaaacbacac
baacbccbaa
cbbaacabcc
baabaacaab

```
bbbbbcacab
acaabcbbba
acaacbccac
cacbabcbba
abcbcbcbcc
ccaccccaba
bcbcabbcca
baabacabca
caababacac
bacacaccaa
bcacbbbbca
bcbbbcaccc
ccabbcccbb
bccbcabbca
baacbabcbc
ccacabccab
caacbcaaab
cacbaaccac
aaccbcaabb
abacabcaab
bacbcbcaca
```

```
Complete the following string:
caacb
```

## H.2.11   RED-GREEN$_S$

You are outside a house of rooms and portals. There are 3 rooms in the house, and each room has 3 unique
      portals labeled a, b, and c. Each portal teleports you to one room of the house (and sometimes the
      destination is the room the portal is in). Every portal in a given room always behaves the same way.

In this house, each of the rooms look exactly the same, except some of the rooms have red walls and some have
      green walls. However, there are *3* rooms in total, so you cannot determine which room you are in by
      color alone, and two rooms of the same color may have portals that behave differently. You've been into
      this house many times before. Each time, as you move through the house, you write down what series of
      portals you take and the color of the room you end up in. You have a collection of paths you've taken
      where you've ended up in a room with green walls, listed below. Given the final incomplete path at the
      bottom, write a series of up to 5 remaining steps that will cause you to end up in a room with green
      walls again.

Tag your final answer like <answer>ab</answer>.

Given these paths that end in a room with green walls:
```
cbcbabbcca
abcaaacbaa
aabccbabbb
bbbccbbbca
aababaccba
aaaacbacac
baacbccbaa
cbbaacabcc
baabaacaab
bbbbbcacab
acaabcbbba
acaacbccac
cacbabcbba
abcbcbcbcc
ccaccccaba
bcbcabbcca
baabacabca
caababacac
bacacaccaa
bcacbbbbca
bcbbbcaccc
ccabbcccbb
bccbcabbca
baacbabcbc
ccacabccab
caacbcaaab
cacbaaccac
aaccbcaabb
abacabcaab
bacbcbcaca
```

```
Complete the following path:
caacb
```

# I    CASE STUDY: SUM MODULO 3 DFA

We investigate the transducer task on the DFA depicted in Figure 1. This DFA can be interpreted as an arithmetic check, where a represents 0, b represents 1, and c represents 2, and the DFA accepts strings whose sum is equal to 0 modulo 3. For this case study, we focus on the model/prompt combinations MB (mistral-nemo-minitron-8B/BASIC$_T$: the best performing non-reasoning model combination overall) and CR (claude-3.5/RED-GREEN$_T$: the best performing non-reasoning model combination that provides an explanation, needed later for our qualitative analysis).

Figure 8a depicts the number of errors each model receives on 1000 instances of the transducer task for this DFA. Nearly all errors made by the 6-GRAM$_T$ model were also made by at least one LLM, while the two LLMs often made unique errors. While this task is better-known than most DFAs, all 3 models perform worse on this DFA than average.

We also performed a qualitative analysis, investigating CR's outputs on the RED-GREEN$_T$ prompt to see what kind of reasoning it is using; specifically we sampled 30 examples where it had the correct answer, and 30 examples where it had the incorrect answer but the 6-GRAM$_T$

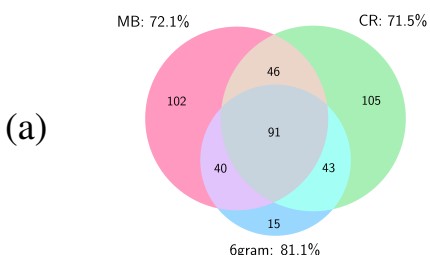

(a)

(b)

|  | Correct | Incorrect |
|---|---|---|
| Total | 100% | 100% |
| a is no-op | 70% | 73% |
| 1b and 1c lead to 0 | 47% | 57% |
| 2-periodic | 30% | 47% |
| 3-periodic | 13% | 13% |
| 2 red rooms | 7% | 10% |

Figure 8:    Results on Sum Modulo 3 DFA. (a)    MB=mistral-nemo-minitron-8B/BASIC$_T$, CR=claude-3.5/RED-GREEN$_T$.    Venn diagram of errors (out of 1000). Labeled percentages are accuracies. (b) Results of qualitative analysis, out of 30 in both cases.

model had the correct answer. Results of this analysis can be found in Figure 8b. We find that, in general, CR is following a 3-GRAM approach, learning rules relating to the conditions under which the previous output and symbol can be used to predict the next output. Specifically, it is able to learn that a does not change the output, and that b and c will lead a 1 state to a 0 state. These results comport with the overall finding of Table 1, where we found that 3-GRAM$_T$ was the largest $n$-GRAM$_T$ that any non-reasoning LLM outperformed, as well as our finding that LLM performance decreases for tasks that are not solvable by $n$-GRAMs; see Appendix B for details.

The model also attempts to identify periodic patterns, but identifies period-2 patterns more than period-3 patterns, despite knowing that there are three "rooms" (states). At no point in any of the 60 reasoning traces analyzed does it realize that this is a version of the Sum Modulo 3 DFA[5], but it does show some glimmers of world modeling: in a few cases it correctly determines that there are two red rooms; however, this does not lead to further discoveries. It is not superior reasoning that leads to correct solutions, rather the correct examples are more likely to be ones that a 3-GRAM model would infer correctly, i.e., those traces ending in a, 1b, or 1c, which occur cumulatively in $\frac{5}{9}$ of cases[6].

Despite transformers' high computational capacity, without the ability to pattern match to existing problems, Claude uses an unsophisticated and ineffectual approach.

---

[5]In fact, in none of the 1000 traces do the substrings "sum" or "mod" appear, except as a part of "assuming"

[6]On the $\sim\frac{5}{9}$ of examples following this pattern, CR achieves 93.5%, to the 6-GRAM$_T$'s 97.3%, and on the remaining $\sim\frac{4}{9}$, it achieves 43.8%, to the 6-GRAM$_T$'s 60.7%. Detailed Venn diagrams on these conditions can be found in Figure 9.

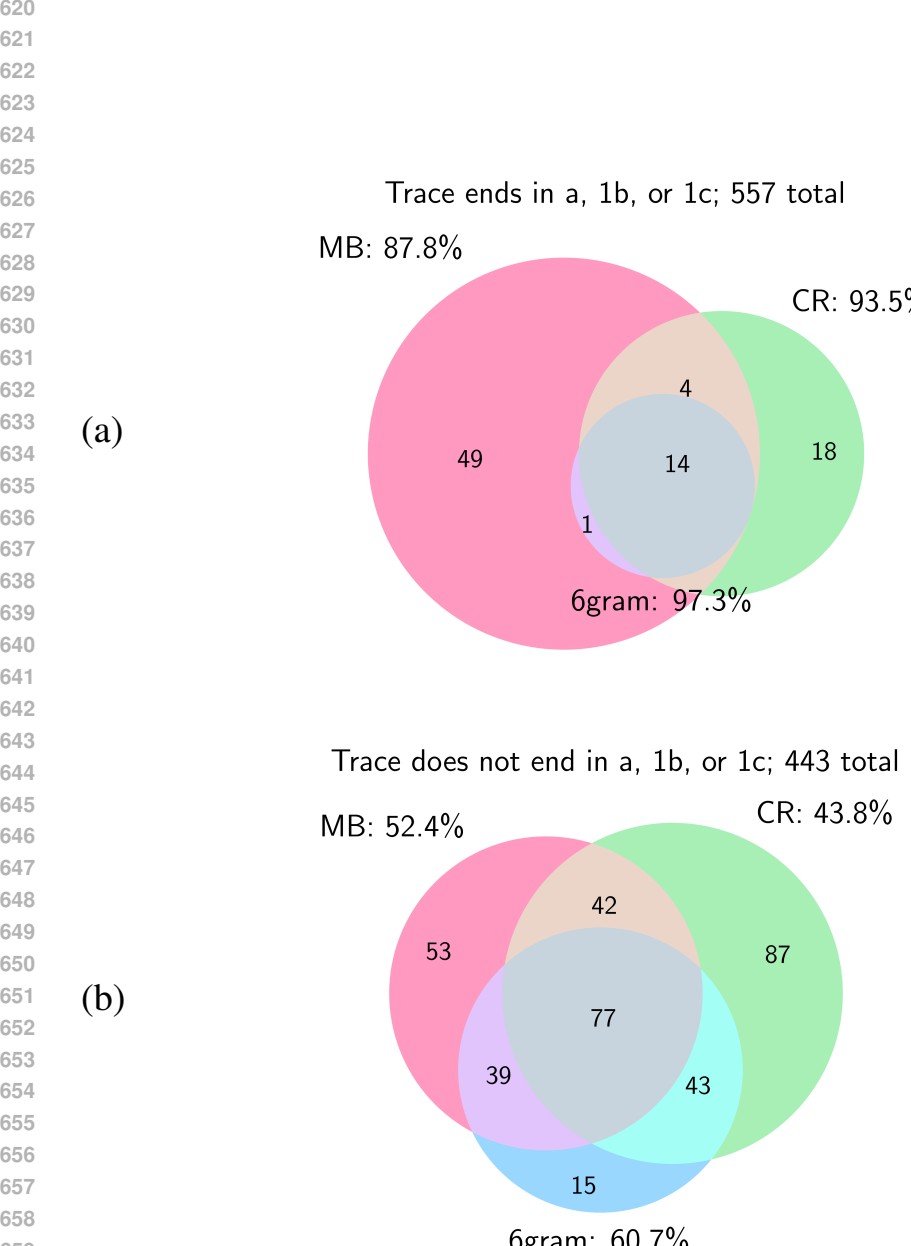

Figure 9: Results on Sum Modulo 3 DFA under trivial / nontrivial conditions. Percentages are accuracy numbers, and venn diagram is error counts. (a) In this condition, CR and the 6-GRAM$_T$ both get very high accuracies, with nearly all 6-GRAM$_T$ also being CR errors. MB does relatively poorly. (b) In this condition, models do significantly more poorly overall, with CR in particular performing worse than chance. Here, errors are more symmetric, with more 6-GRAM$_T$ errors that are not accounted for by either or both model, indicating that a larger fraction of both successes and failures in this condition are down to random chance.

## J    COMPUTE USAGE

The experiments in this paper on proprietary models had the following (approximate) costs.

- gpt-5: $920
- o3-mini: $430
- 4o: somewhere between $100 and $200
- 4o-mini: somewhere between $50 and $150
- claude-3.5: $80

The open weight experiments took a cumulative 10-50 GPU-hours on NVIDIA RTX 6000 Ada Generation GPUs, some models requireed the use of 4 in parallel.

# K DIFFERENT SYMBOLS

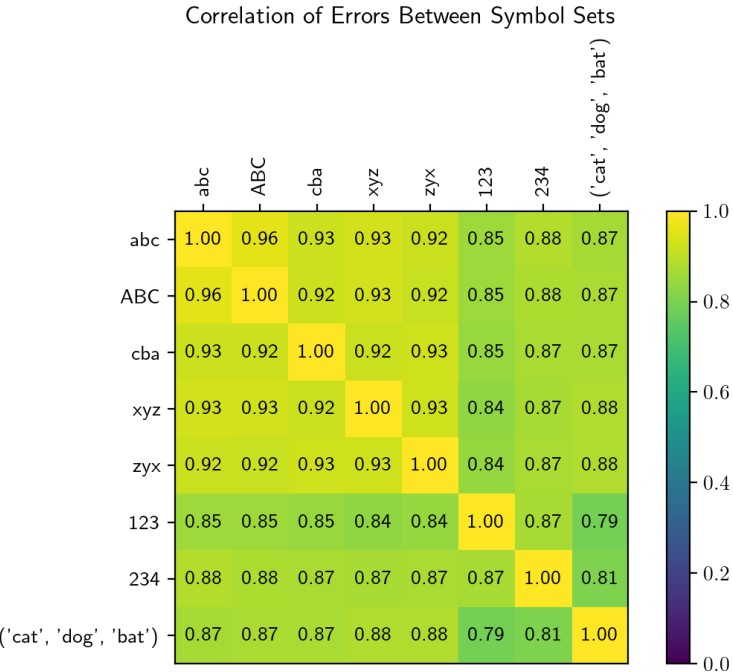

Figure 10: Correlations between different versions of the same prompt on BASIC$_T$ prompt, and the `mistral-nemo-minitron-8B`. abc is the original prompt, xyz reflects the same problems but with x instead of a.

To verify that our experiments are not particularly sensitive to the choice of input symbols abc, we perform a series of experiments in which we replace these symbols with others. For this experiment, we use the BASIC$_T$ prompt, and the `mistral-nemo-minitron-8B` model (chosen to match Appendix C). All of these settings still underperform 4-GRAM$_T$. We run all experiments for 1000 different DFAs, and compute correlations between each of the elements.

Figure 10 shows the relationship between different symbol sets. We fid that all the letter-based combinations correlate heavily with each other, with the number-based ones correlating less well, and cat/dog/bat having poor correlations with both other categories, particularly the numeric ones. In general, we find strong alignment, indicating that the models are not particularly relevant to the specific iput symbols and are instead treating the input appropriately symbolically.

## L  SEQUENCE COMPLETION: NOVELTY BY CORRECTNESS

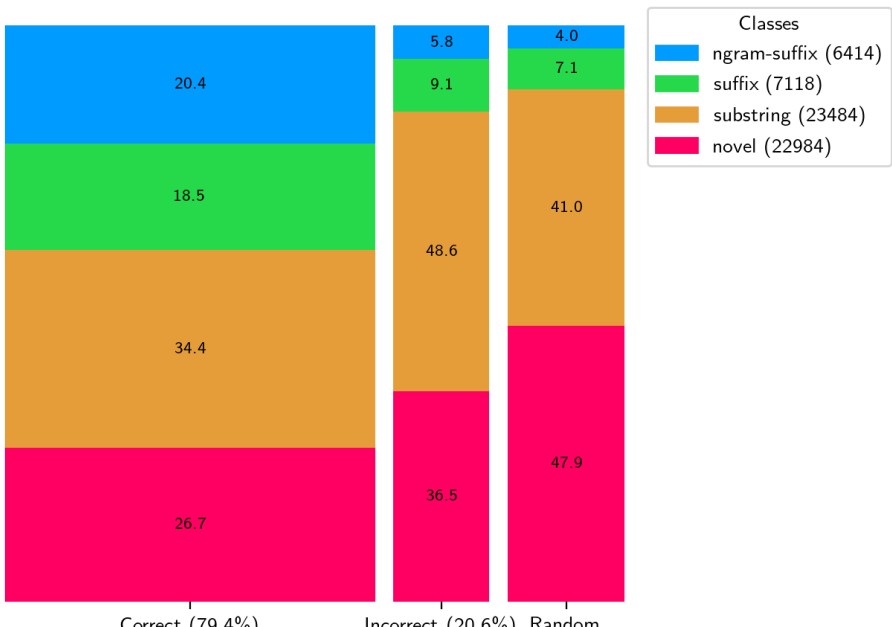

Figure 11: The number of elements of each completion class, separated out by whether it is correct/incorrect. Each column's width is proportional to the number of elements in it, except the Random control. Numbers indicate the percentage of elements in that column within that particular cell, e.g., 20.4% of correct completions are an n-gram-like suffix.

To investigate how much LLMs are operating in sequence completion, we run the best open source model (`qwen-2.5-coder-7B`) on 1000 examples, and then classify the completions as either

1. an n-gram-like suffix (i.e., the last character of the prefix + the generated completion appears as a suffix in one of the given examples)
2. a suffix of one of the given examples
3. a substring of one of the given examples
4. an entirely novel string

We find (Figure 11) that while there is some bias towards suffixes, particularly in completions that end up being correct, there are quite a few novel completions, and in fact these represent about 27% of correct completions. As such, we can see that the model is not quite running an n-gram in the sequence completion task. However, it is possible that many of these are only correct by chance, as on average there is a 53% chance that any random string will be correct on this dataset (as seen in the RANDOM$_S$ baseline; see Table 1).

