# OpenReview forum: "Randomly Sampled Language Reasoning Problems Elucidate Limitations of In-Context Learning"
_ICLR.cc/2026/Conference — ICLR 2026 Conference Withdrawn Submission_

### Official Review · Reviewer_8EWz · 2025-10-20

**Soundness:** 2
**Presentation:** 3
**Contribution:** 1
**Rating:** 2
**Confidence:** 4

**Summary:**

The paper evaluates the in-context learning (ICL) capability of large language models (LLMs) on tasks pertaining to regular languages recognized by 3-state deterministic finite automata (DFAs). The main result is that certain n-gram models outperform LLMs on sequence completion and transduction tasks on such regular languages. Overall, the paper extends previous works on in-context learning of regular languages to pre-trained LLMs, but suffers from major concerns regarding claims and contributions.

**Strengths:**

1. The presentation of the experiments and results is clear without any obvious/major issues.

2. The evaluations cover a wide variety of models, including open-weights, open-code, and proprietary models.

**Weaknesses:**

1. The novelty of the findings in this paper is rather limited and does not justify the claims made in Section 1 (contributions). In particular, the authors claim that they introduce an LLM ICL benchmark for novel tasks using the regular languages, but do not justify the motivation for such a new benchmark when compared to existing works, such as RegBench in [1]. Also, no experiment/result discusses the effects of RLHF on the ICL performance. I would suggest a rephrasing of the claims to avoid such confusion.

2. The transductive task can also be treated as a symbolic reasoning task without associating any language with it. The task considered in this paper is to check if there is an even number of a's" in the string or not. Similar character counting tasks have been explored in [2] as well (which can be treated as a variant of the prompt where the task is revealed in the instruction). To this end, I believe that the experiments in this work are rather incremental and lack a strong motivation.

I am willing to engage in a discussion and better understand the author's perspective on this matter. Furthermore, since an LLM has to be trained on multi-lingual data for solving multi-lingual tasks, it is evident in the literature that it is not learning a general theory of language. Also, because the transductive tasks can be framed as symbolic reasoning tasks, I am not convinced of the importance of the current results and the importance of framing such tasks as "random language tasks".

[1] Akyürek, Ekin, et al. "In-Context Language Learning: Architectures and Algorithms." International Conference on Machine Learning. PMLR, 2024.

[2] Shin, Andrew, and Kunitake Kaneko. "Large Language Models Lack Understanding of Character Composition of Words." ICML 2024 Workshop on LLMs and Cognition.

**Questions:**

1. **see weaknesses for the major questions.**

2. What can be some future research directions based on this work?

3. Sampling a string from a DFA can also be framed as sampling from a markov chain. The work by [3] explores this problem of learning such inherent structures as well. Can the authors discuss the similarities/tradeoffs with such works since they are closely related?

4. How does the cardinality of states of a DFA affect performance? As of now the paper considers 3 states but say without the brute-force baseline and closed-soiurce models (which maybe expensive),  if we increase the state count to 4,5,6 etc, do we see a pattern in open-weights/code LLM performance?

minor: recheck grammar. For example in section 2.3 (second paragrah: "We also LLM ICL but push ...".


[3] Edelman, Ezra, et al. "The evolution of statistical induction heads: In-context learning markov chains." Advances in neural information processing systems 37 (2024): 64273-64311.

---

> ### Author Response · Authors · 2025-11-19
>
> Thank you for your review! We have provided responses to your questions in two parts.
>
> > The novelty of the findings in this paper is rather limited and does not justify the claims made in Section 1 (contributions). In particular, the authors claim that they introduce an LLM ICL benchmark for novel tasks using the regular languages, but do not justify the motivation for such a new benchmark when compared to existing works, such as RegBench in [1].
>
> We believe that our benchmark is different from that of RegBench in form, but much more significantly, in how it is used in these experiments. RegBench is a benchmark used to train transformers in ICL, whereas our benchmark is used to evaluate LLM ICL.
>
> This difference is massive as we are answering entirely separate questions: RegBench is answering “what are the limits of the capacity of small transformers to learn in-context" whereas we are asking "how well have massive transformers pre trained on massive datasets learned the ability to emergently solve novel problems using ICL”.
>
> As such, the forms of these benchmarks are different, with RegBench being focused on stochastic languages corresponding to 4-12 states and us focusing on far smaller examples. We have a discussion of [1] in our related work section.
>
> > Also, no experiment/result discusses the effects of RLHF on the ICL performance. I would suggest a rephrasing of the claims to avoid such confusion.
>
> Table 1 annotates which models are instruction tuned, and we have a brief discussion of this in Section 5.2. However, you are correct that this is not a central enough focus of our analysis to warrant its place in the contributions, so we have deemphasized it.
>
> > The transductive task can also be treated as a symbolic reasoning task without associating any language with it. The task considered in this paper is to check if there is an even number of a's" in the string or not. Similar character counting tasks have been explored in [2] as well (which can be treated as a variant of the prompt where the task is revealed in the instruction). To this end, I believe that the experiments in this work are rather incremental and lack a strong motivation.
> > I am willing to engage in a discussion and better understand the author's perspective on this matter. Furthermore, since an LLM has to be trained on multi-lingual data for solving multi-lingual tasks, it is evident in the literature that it is not learning a general theory of language. Also, because the transductive tasks can be framed as symbolic reasoning tasks, I am not convinced of the importance of the current results and the importance of framing such tasks as "random language tasks".
>
> In the statistical language modeling community, the word “language” is generally used to refer to a set of strings defined by some grammar. This encapsulates both natural languages and formal ones. Many common use cases of LLMs require them to produce output that adheres to a particular format, where the LLM learns that format from in context examples. This is extensively done in practice, and conventional wisdom suggests that it works fairly well. However, it is difficult to tell whether this is because models are good at learning the format from the examples (which formally speaking, is a language learning problem, though different from learning e.g., English) or whether it is because the examples guide the model towards parametric knowledge learned from the training dataset. In this work, we use the simplest kind of nontrivial formal language (regular languages corresponding to 3 state DFAs) to probe at the ability of LLMs to learn the format from examples in a context where there is no relevant knowledge from the training data to be guided towards.
>
> This task is completely different from the one in [2] which is primarily about subtokenization, a topic that we avoid by not using characters that compose into coherent words. In particular for the transducer task, all characters are separated by delimiters, and for Sequence Completion, adding delimiters reduces performance, demonstrating low performance is not a tokenization artifact (see Appendix for more details).

---

> > ### Author Response · Authors · 2025-11-19
> >
> > > What can be some future research directions based on this work?
> >
> > There are a variety of interesting research questions that we believe this paper raises. First: what specifically are LLMs doing when encountering an unstructured language reasoning problem like this? What algorithms are they applying? Secondly, is it possible to train an LLM on formal language data in such a way that it is capable of some kind of general language understanding, and can operate well on all kinds of formal languages, or is this beyond the capacity of these networks? Other research questions that might be interesting involve weakening the unfamiliarity of the domain and seeing what happens; we do so by giving away the problem structure and find that only GPT-5 is able to really exploit this; but it is possible that other pieces of information might help more models solve this task.
> >
> > > Sampling a string from a DFA can also be framed as sampling from a markov chain. The work by [3] explores this problem of learning such inherent structures as well. Can the authors discuss the similarities/tradeoffs with such works since they are closely related?
> >
> > This paper appears to be closely related to Ayurek 2024 which we discuss in related work. In both cases, the work focuses on training relatively small transformers on the specific ICL task being considered, in order to evaluate model capacity in general. We are interested in very large transformers that are said to have emergent capabilities, and wish to determine whether the training process for these models has produced an artifact that can solve these ICL tasks out of the box. We have added a discussion of this paper to related work.
> >
> > > How does the cardinality of states of a DFA affect performance? As of now the paper considers 3 states but say without the brute-force baseline and closed-soiurce models (which maybe expensive), if we increase the state count to 4,5,6 etc, do we see a pattern in open-weights/code LLM performance?
> >
> > We have performed the experiment with more states, with results in Appendix C.2, using the same model as the length experiment (the nemo-minitron-8B model was chosen because it is statistically tied for first place in transducer results with qwen-2.5-7B but much better on sequence completion). We find that in general, both the LLM and the n-gram models decrease in performance as the number of DFA states increases, about the same amount, so interpretation of the results does not change.
> >
> > > minor: recheck grammar. For example in section 2.3 (second paragrah: "We also LLM ICL but push ...".
> >
> > Thanks for pointing this out, fixed!

---

> > > ### Comment · Reviewer_8EWz · 2025-11-20
> > >
> > > Thanks for the response. Please find the follow-up questions below.
> > >
> > > > We believe that our benchmark is different from that of RegBench in form, but much more significantly, in how it is used in these experiments. RegBench is a benchmark used to train transformers in ICL, whereas our benchmark is used to evaluate LLM ICL. This difference is massive as we are answering entirely separate questions: RegBench is answering “what are the limits of the capacity of small transformers to learn in-context" whereas we are asking "how well have massive transformers pre trained on massive datasets learned the ability to emergently solve novel problems using ICL”.
> > >
> > > Q1: Why not just use RegBench for the ICL experiments? I understand that the authors of that paper used it for training and eval but the tasks are still rich enough to be used for inference only with modern LLMs and compared to the n-gram baselines. The setup/dataset of the current paper does not seem to be much richer or complex than RegBench as well (as the authors have stated).
> > >
> > > Q2: Secondly, I think Figure 5 in Appendix C.1 misses a point. Although the model performance seems to improve as number of examples increases, the curve can look significantly different for larger models.  So, in a way, we are missing out on a key observation that can essentially clarify how model size and number of examples beat the baselines. Without this observation, I think the claim about "significant weaknesses in large language models’ ability to in-context-learn entirely novel language reasoning problems" in conclusion is not justified. Furthermore, also note that since we are anyway dealing with custom tasks with only a few tokens representing each in-context example, thousands of examples here should not be compared with typical NLP style in-context examples (as long as the overall sequence length is reasonably long and supported by the model).
> > >
> > > Q3. In essence, the claims about the models not learning or matching a specific algorithm are also not valid since they act as function approximators in context and do not actually mimic the computation itself. So, unless there is evidence for model performance with scaling examples and model size (not necessarily 100B but maybe a curve with 3B, 7B, 14B, 32B etc or similar scales), the claims about function approximation are also not justified.

---

> > > > ### Author Response · Authors · 2025-11-21
> > > >
> > > > In response to Q1:
> > > >
> > > > There are two primary reasons we did not pick RegBench. Firstly, RegBench is a much more difficult benchmark than what we set up. It involves more states (4-12), and generates PDFAs, which create stochastic rather than regular languages. We wish to investigate relatively easy language problems, and as such decided to construct our own transducer benchmark set. Secondly, RegBench has a few quirks in the sampling procedure that are relevant within the context of the stochastic languages, but not in our case. E.g., the special state S_0 and how it is handled in transitions (see footnote 3 in Akyürek 2024). Finally, RegBench does not have a sequence completion mode; it is unclear how the techniques designed to alter the difficulty of RegBench would generalize. All in all, we decided a simpler, more parsimonious, sampling approach would be better for our paper.
> > > >
> > > > In response to Q2 and Q3:
> > > >
> > > > To ensure the results we are getting as a function of distance are not unique to small models, we ran the same experiment for the best performing 32B parameter and 70B parameter models, receiving qualitatively similar results (increasing performance with examples, not catching up to the best n-Gram). See the updated Figure 5 for the detailed results of our experiment.
> > > >
> > > > Additionally, though the examples are generally smaller (really, more compressible, as they overlap, technically in the Transducer task each “example” is the entire prefix, making the mean example length in the 1200 example task be 600 pairs of tokens long), the number of examples is still quite large. This provides a huge amount of information to the model, which would not be the case in a typical ICL use case. As such, we do not think it is a significant limitation that the n-Gram models saturate performance by 1200 examples.

---

> ### Comment · Reviewer_8EWz · 2025-11-27
>
> Comparing models of different sizes, especially when they belong to different model families, is unfair. Qwen-2.5 is a relatively newer model when compared to llama-3-70B. The pre- and post-training differ significantly, and the performance difference cannot be attributed solely to model size. This sends a wrong message to the readers that the model size does not matter and can also hurt performance (as per Figure 6).
>
> I thank the authors for their responses, but overall, I am still not convinced about the utility of the benchmark and the insights. Most importantly, there is currently no way to systematically analyze which factors of LLM design and training approaches affect the benchmark scores (see also the point above, as I am mainly concerned about wrong takeaways from this work). I will retain the score.

---

### Official Review · Reviewer_EtLQ · 2025-10-30

**Soundness:** 3
**Presentation:** 2
**Contribution:** 2
**Rating:** 2
**Confidence:** 4

**Summary:**

The authors study the in-context learning ability of large language models through the lens of regular languages, in particular whether these models can complete sequences or answer questions about them (sequence completion vs transduction tasks). A large scale study is conducted on regular expressions accepted by 3-state deterministic finite automata, where each LLM gets as input a sequence of examples that are accepted by a randomly drawn automata and then a partial sequence for which it has to provide a completion. The results indicate that while LLMs are able to solve a lot of novel tasks at inference and provide complex reasonings, they are still behind in terms of general purpose meta-learning and currently lag behind even n-gram methods when it comes to regular languages.

**Strengths:**

- The authors study a clear hypothesis of whether current LLMs act as general-purpose in-context learners and highlight through a study with regular languages that they are not; thereby clearly validating their hypothesis. In fact, it is a clear study with a testable hypothesis and the authors succinctly provide a conclusion to the question that they ask.
- The evaluation conducted is quite thorough from the lens of the number of DFAs and examples used to evaluate the models as well as the various different LLMs used, of varying families, sizes and capabilities.

**Weaknesses:**

- While I commend the authors on the clear, well-studied and scoped-out problem formulation, I struggle to grasp the more general benefit behind such an analysis. What the authors show is a clear existence of a problem which can be templated within the language domain on which LLMs struggle. However, a discussion along the lines of *no free lunch* would have been quite helpful, in fact it is not unbelievable that such models would struggle on a lot of language-based tasks which are, in some sense, not well encapsulated within their pre-training methodology. The benefit of ICL is not that they can solve *all* possible problems but that they can solve a number of novel problems that are of interest to us. In fact, if one was to think of ICL as a learning mechanism one can come up with tasks where a general learner like SGD would fail; the benefit of SGD (or ICL) is that it is just generally good at tasks of interest to us.
- There was a lack of analysis into task complexity, in particular from the draft we know that LLMs struggle with regular languages in-context and we also know that in general language tasks they have shown tremendous progress. It would have been nice to see some analysis on how such models behave going from Type 3 to Type 0 languages, and also within Type 3 language going to a small state DFA to a larger state DFA.
- Given that there are no consistent trends between model size and performance, whether it be across or within a model family, it suggests that the phenomena the authors are seeing is a consequence of just the evaluation task being quite out-of-distribution from the training regime. The lack of trends between Basic and CoT prompting, among other variants, also reinforces this hypothesis. In light of this, I would appreciate if the authors could comment on the downstream usefulness of their finding, except highlighting just the existence of a task on which LLMs fail?

**Questions:**

- LLMs are in general somewhat sensitive to how the prompts are formalized as well as the overall template in which observations are provided. From that perspective, have the authors done some form of sensitivity analysis where the characters a/b/c are replaced by different random words or tokens? For example a/b/c could be replaced by 0/1/2 or cat/dog/bat. It would be important to understand if ICL even learns this invariance from in-context observations, and understand its implications on the current hypothesis.

---

> ### Author Response · Authors · 2025-11-19
>
> Thank you for your review! We have provided a response in two parts.
>
> > While I commend the authors on the clear, well-studied and scoped-out problem formulation, I struggle to grasp the more general benefit behind such an analysis. What the authors show is a clear existence of a problem which can be templated within the language domain on which LLMs struggle. However, a discussion along the lines of no free lunch would have been quite helpful, in fact it is not unbelievable that such models would struggle on a lot of language-based tasks which are, in some sense, not well encapsulated within their pre-training methodology. The benefit of ICL is not that they can solve all possible problems but that they can solve a number of novel problems that are of interest to us. In fact, if one was to think of ICL as a learning mechanism one can come up with tasks where a general learner like SGD would fail; the benefit of SGD (or ICL) is that it is just generally good at tasks of interest to us.
> > Given that there are no consistent trends between model size and performance, whether it be across or within a model family, it suggests that the phenomena the authors are seeing is a consequence of just the evaluation task being quite out-of-distribution from the training regime. The lack of trends between Basic and CoT prompting, among other variants, also reinforces this hypothesis. In light of this, I would appreciate if the authors could comment on the downstream usefulness of their finding, except highlighting just the existence of a task on which LLMs fail?
>
> While it is true that all algorithms have specific failure modes, we think that it is important to understand what these failure modes are. In this paper, we analyze performance on a benchmark that does not have the hallmarks of a  difficult problem, except in its task novelty. The fact that LLM ICL performs quite poorly on this task is, while not unbelievable, still an important finding to provide relatively definitive evidence for.
>
> We strongly dispute the implicit claim that it is an uncontroversial and obvious view that LLM ICL performs poorly on tasks that are dissimilar from those in the training data of the LLM. The belief that LLM ICL can emergently solve novel problems is an extremely common belief both in the academic literature (e.g., outperforming finetuning [1], explanations involving ICL being a form of gradient descent [2], it can “handle various unseen tasks” [3], and it can handle novel DSLs [4]) and in industry, where many leaders claim that LLM ICL will replace much of significant economic activity within the next few years.
>
> Additionally, we find evidence that ICL is in fact acting as a learning algorithm on our particular task. While it is true that results across different model families are scattershot, we demonstrate in particular in Appendix C that increasing the number of examples causes LLM performance to rise, in fact faster than individual n-grams (though always remaining worse than the best available n-gram). In addition, we find that increasing task difficulty (by increasing the number of states) causes LLMs to reduce in accuracy, similar to n-grams [see below for greater discussion]. As such, we conclude not that ICL is useless or not a learning algorithm, rather that it is significantly more limited on random examples than many people think, and does not appear to be taking advantage of model scale in the way that people imagine.
>
> Finally, we do not believe that our tasks are that different from the kinds of tasks that many people are interested in, except in their simplicity. Fundamentally, Sequence Completion and Transducer correspond to generation and classification tasks, which are commonly used. The fact that we are using randomly sampled languages simulates the fact that novel problems often do not have direct precedent in existing data. We believe that it is not unreasonable to evaluate LLMs on this kind of task.
>
> [1] Yin, Qingyu, et al. "Deeper insights without updates: The power of in-context learning over fine-tuning." Findings of the Association for Computational Linguistics: EMNLP 2024. 2024.
> [2] Dai, Damai, et al. "Why can GPT learn in-context? language models secretly perform gradient descent as meta-optimizers." Findings of the Association for Computational Linguistics: ACL 2023. 2023.
> [3] Wang, Fan, et al. "Benchmarking General-Purpose In-Context Learning." arXiv preprint arXiv:2405.17234 (2024).
> [4] Bogin, Ben, et al. "Leveraging code to improve in-context learning for semantic parsing." Proceedings of the 2024 Conference of the North American Chapter of the Association for Computational Linguistics: Human Language Technologies (Volume 1: Long Papers). 2024.

---

> > ### Author Response · Authors · 2025-11-19
> >
> > > There was a lack of analysis into task complexity, in particular from the draft we know that LLMs struggle with regular languages in-context and we also know that in general language tasks they have shown tremendous progress. It would have been nice to see some analysis on how such models behave going from Type 3 to Type 0 languages, and also within Type 3 language going to a small state DFA to a larger state DFA.
> >
> > We have performed the experiment with more states, with results in Appendix C.2, using the same model as the length experiment (the nemo-minitron-8B model was chosen because it is statistically tied for first place in transducer results with qwen-2.5-7B but much better on sequence completion). We find that in general, both the LLM and the n-gram models decrease in performance as the number of DFA states increases, about the same amount, so interpretation of the results does not change.
> >
> > On the other hand, we do not believe that there is any coherent way to compare to more complicated families of languages; fundamentally it would require the ability to sample “similarly" across different language families, but they all have different parameterizations and therefore would be potentially much easier or much harder even if using a superficially simple sampling technique.
> >
> > > LLMs are in general somewhat sensitive to how the prompts are formalized as well as the overall template in which observations are provided. From that perspective, have the authors done some form of sensitivity analysis where the characters a/b/c are replaced by different random words or tokens? For example a/b/c could be replaced by 0/1/2 or cat/dog/bat. It would be important to understand if ICL even learns this invariance from in-context observations, and understand its implications on the current hypothesis
> >
> > Thank you for your suggestion! We have performed this experiment and found that in all cases we analyzed (including 1/2/3 and cat/dog/bat) the results on 1000 samples correlated strongly with a/b/c, with the strongest correlations being alternate letterings and weakest being cat/dog/bat vs 1/2/3. We also tried replacing a/b/c with c/b/a (i.e., swapping c and a). [We also tried 0/1/2 but this reduced accuracy, my suspicion is this is because the 0 and 1 from the outputs were being confused with the 0 and 1 from the symbols in the long list of interleaved inputs/outputs].
> >
> > All results were at topline similar, and all ended up correlating with the a/b/c results with r>0.85. See Appendix K in the updated paper for details.
> >
> > As a result, we believe that our experiment is not particularly sensitive to the choice of symbols; a result that provides additional credence to the idea that what is going on here is some true learning algorithm that is simply worse than n-Grams.

---

> > > ### Comment · Reviewer_EtLQ · 2025-11-27
> > >
> > > Thank you for your response. While I can see the value in this work as a proposed dataset and consequent study of how LLMs fare with regular languages, I am not convinced about the utility of the tasks used. In particular, one could design a number of different tasks on which LLMs and their ability to perform implicit optimization would be suboptimal compared to specific algorithms (eg. one could design HMMs, among other tasks).
> > >
> > > This work needs some notion of difficulty of tasks (number of states, kind of language, etc.) to better and more cleanly understand how different models perform, and then it could serve as a better benchmark.

---

### Official Review · Reviewer_AyAV · 2025-10-31

**Soundness:** 3
**Presentation:** 3
**Contribution:** 2
**Rating:** 6
**Confidence:** 3

**Summary:**

The paper investigates whether large language models (LLMs) are genuinely capable of learning new tasks in an in-context learning (ICL) setting. To study this question, the authors design controlled experiments based on simple formal language tasks generated from random deterministic finite automata (DFAs) with three states and three symbols. They introduce two tasks: (1) sequence completion, where the model must complete a query prefix consistent with in-context examples of a random DFA, and (2) transduction, where the model must predict the *transducer* output corresponding to an annotated input sequence drawn from a DFA.

The authors evaluate a broad range of pretrained open-weight and proprietary LLMs under different prompting strategies, from basic prompts with minimal explanation accompanying the in-context examples to more elaborate chain-of-thought (CoT) prompts. They compare these results against several non-LLM baselines, most notably simple n-gram models that simply match query patterns directly to the in-context examples.

Empirically, they report that the n-gram baseline consistently outperforms the LLMs across both tasks, and CoT prompting fails to produce systematic improvements, highlighting a fundamental limitation of current LLMs in performing genuine novel task learning through the ICL framework.

**Strengths:**

The paper addresses an interesting question about the nature of in-context learning in LLMs. The setup is well-motivated, accompanied by a thorough related work section. The authors justify their benchmark design choices well and provide clear reasoning behind each task setup.

The experimental evaluation also covers a wide range of pretrained LLMs, along with detailed reporting of implementation, prompting formats, and compute usage. The additional ablations, such as varying the number of in-context examples and analyzing the effects of tokenization, add more depth to the empirical analysis.

**Weaknesses:**

(See questions)

**Questions:**

1. **On the complexity of the task**: The authors motivate their design choice of using simple DFAs, but I am concerned that the task may be too simple to meaningfully assess whether LLMs learn the underlying grammar. Mainly, the n-gram baselines achieve over 90% accuracy, indicating that simple pattern copying can result in a near-perfect performance on this task, without any knowledge of the grammar structure. Isn't this a weakness of this benchmark for assessing the model's capability in capturing "world model"?

2. **Number of in-context examples**: It may be the case that LLMs can learn the task in-context, but are not sample efficient. If so, providing only a few in-context examples might underestimate their ability to pick up the underlying transition patterns.

    Figure 5, which explores the impact of increasing the number of examples, partially addresses this. However, since LLM performance improves much more sharply than the n-gram baseline as the number of examples grows, I wonder whether LLMs could eventually catch up with enough in-context data.

3. **Novelty of the completions**: Other than generating a valid output/completion, another performance metric for uncovering the ICL mechanism is to check the novelty of the generated outputs themselves: When a model produces a correct completion, is it simply copying a pattern present in the in-context examples, or is it generating a new suffix consistent with the underlying grammar?

    One possible diagnostic would be to compare the valid LLM completions with those pattern(s) matched by the n-gram model to check whether the LLM is reproducing existing context patterns or generating genuinely new suffixes that reflect partial inferred structure (partial, since the performance is not perfect).

4. **Line 478**: ``We believe our results suggest that LLMs have learned individual models of particular languages, but not a general theory of language.'' Could you clarify this statement? What does it mean that LLMs “have learned individual models of particular languages,” and how is this conclusion supported by the presented results?

5. **Minor**: Could you clarify the *common-suffix* baseline? Does the completion "*always* end in an accept state", or does it refer to the completion that most frequently leads to an accept state across queries?

---

> ### Author Response · Authors · 2025-11-19
>
> Thank you for your review! We have responded to your questions in two parts.
>
> > On the complexity of the task: The authors motivate their design choice of using simple DFAs, but I am concerned that the task may be too simple to meaningfully assess whether LLMs learn the underlying grammar. Mainly, the n-gram baselines achieve over 90% accuracy, indicating that simple pattern copying can result in a near-perfect performance on this task, without any knowledge of the grammar structure. Isn't this a weakness of this benchmark for assessing the model's capability in capturing "world model"?
>
> We specifically want to avoid evaluating the ability of ICL to perform world modeling. World modeling is a complex task that is known to be a point of relatively lower performance for LLMs; and as such we want to exclude it from our analysis as a point of consideration, to explore other potential explanations. Therefore, we are explicitly using n-gram models as baselines to compare against a pure pattern recognition baseline. As for saturated performance, while n-grams do perform well, there is room for statistically significant improvement; e.g., both the BruteForce algorithm and GPT-5 when given the problem structure are able to solve the problem substantially better than 6-Gram.
>
> > Number of in-context examples: It may be the case that LLMs can learn the task in-context, but are not sample efficient. If so, providing only a few in-context examples might underestimate their ability to pick up the underlying transition patterns. Figure 5, which explores the impact of increasing the number of examples, partially addresses this. However, since LLM performance improves much more sharply than the n-gram baseline as the number of examples grows, I wonder whether LLMs could eventually catch up with enough in-context data.
>
> We do not believe that LLM ICL performance will meaningfully catch up with n-gram performance as the number of examples increases. Extending the results to 1200 examples (see updated PDF for updated figure in Appendix C.1), we find that on Sequence completion the LLM continues to improve but does not catch up to 4Gram. On Transducer, the LLM (accuracy 99.06%) catches up with 6gram (98.67%) and almost 7gram (99.12%), but at this point the largest ngram available (which we name ∞-gram) is getting a performance of 99.77%, which is effectively saturated performance. Additionally, at this point, we are well beyond the realm of the number of examples generally used in In-Context-Learning.
>
> > Novelty of the completions: Other than generating a valid output/completion, another performance metric for uncovering the ICL mechanism is to check the novelty of the generated outputs themselves: When a model produces a correct completion, is it simply copying a pattern present in the in-context examples, or is it generating a new suffix consistent with the underlying grammar? One possible diagnostic would be to compare the valid LLM completions with those pattern(s) matched by the n-gram model to check whether the LLM is reproducing existing context patterns or generating genuinely new suffixes that reflect partial inferred structure (partial, since the performance is not perfect).
>
> We performed a version of this analysis, classifying the completions as either
>
> [a] an n-gram-like suffix (i.e., the last character of the prefix + the generated completion appears as a suffix in one of the given examples)
> [b] a suffix of one of the given examples
> [c] a substring of one of the given examples
> [d] an entirely novel string
>
> We find (see the figure in Appendix L) that while there is some bias towards suffixes, particularly in completions that end up being correct, there are quite a few novel completions, and in fact these represent about 27% of correct completions. As such, we can see that the model is not quite running an n-gram in the sequence completion task. However, it is possible that many of these are only correct by chance, as on average there is  a 51% chance that any random string will be correct on this dataset (as seen in the Random Baseline). Due to this issue, it ends up being difficult to determine whether a particular novel completion is a total guess or the result of coherent reasoning. In practice, our results imply that these models are using the kind of sequential copying that our baselines use, but that is not all they are doing; though they are generally more successful when doing copying.

---

> > ### Author Response · Authors · 2025-11-19
> >
> > > Line 478: ``We believe our results suggest that LLMs have learned individual models of particular languages, but not a general theory of language.'' Could you clarify this statement? What does it mean that LLMs “have learned individual models of particular languages,” and how is this conclusion supported by the presented results?
> >
> > We believe LLMs “have learned individual models of particular languages” not because of any of our results but rather from the large body of pre-existing work demonstrating that LLMs excel at a variety of language tasks. This was phrased incorrectly in the original paper, and we have updated the paper to clarify this.
> >
> > > Minor: Could you clarify the common-suffix baseline? Does the completion "always end in an accept state", or does it refer to the completion that most frequently leads to an accept state across queries?
> >
> > The common suffix is always a suffix drawn from the provided examples, that balances length and number of appearances. It always ends in an accept state for the particular strings that have been provided, but is not tested on novel strings, as that is impossible to do without knowing the DFA. The sentence referencing the universal completion was confusing and added little to the description of this baseline, so it has been removed.

---

### Official Review · Reviewer_27f6 · 2025-11-03

**Soundness:** 3
**Presentation:** 3
**Contribution:** 2
**Rating:** 4
**Confidence:** 4

**Summary:**

This paper introduces a benchmark designed to evaluate the in-context learning (ICL) capabilities of large language models (LLMs) in the context of alien (synthetically generated) language reasoning. The authors test several LLMs on this benchmark and find that even state-of-the-art models struggle with even the simplest reasoning tasks in this setting.

**Strengths:**

1.	The paper introduces a novel benchmark that specifically targets and isolates the in-context learning ability of LLMs in a controlled setting involving synthetic or "alien" languages.
2.	The authors provide a comprehensive comparison of multiple LLMs and statistical models, offering a broad overview of current model capabilities on the proposed task.

**Weaknesses:**

1.	While the use of next-token prediction accuracy is straightforward, the paper could benefit from including additional evaluation metrics—such as error type analysis or model calibration measures—to provide deeper insights into the specific failure modes of the models.
2.	The analysis of why LLMs perform well on natural or regular languages but fail on randomly generated ones is limited. A more in-depth investigation into this contrast would strengthen the paper's impact.

**Questions:**

1.	What key insights does this benchmark provide regarding the limitations of current LLMs in in-context learning? And what potential directions could address these limitations?

---

> ### Author Response · Authors · 2025-11-19
>
> Thank you for your review! Responses to individual questions are presented below
>
> > While the use of next-token prediction accuracy is straightforward, the paper could benefit from including additional evaluation metrics—such as error type analysis or model calibration measures—to provide deeper insights into the specific failure modes of the models.
>
> We provide an analysis of the error rates relative to problem difficulty in Appendix B; specifically, these results demonstrate that all models have performance that reduces monotonically as the minimum n-gram needed to solve a particular task grows. This implies that the models are running an algorithm similar to that of n-grams; results from Table 1 show that this is less effective. This is further corroborated by our case study, found in Appendix I, which shows that at least claude 3.5 is running what is effectively a noisy 4-gram model.
>
> > The analysis of why LLMs perform well on natural or regular languages but fail on randomly generated ones is limited. A more in-depth investigation into this contrast would strengthen the paper's impact.
>
> We believe that what is happening here is not directly related to natural languages; rather the difference is between languages that are similar to ones found in the dataset and languages that are more dissimilar (which randomness induces).
>
> The specific mechanism we have in mind for why this might be the case is the notion of an LLM as a tool that identifies a given prompt as belonging to some known bin of problems and then applies an algorithm it has learned during training to solve that problem. Our main line of evidence to demonstrate this hypothesis is that GPT5 has unremarkable performance on the Transducer task, coming in 15th place, however when given a prompt that contains the problem structure (DFA-COT or Red-Green), it is able to perfectly solve the task as well as Brute-Force. The task is identical, with the only difference being that in one case it has the problem bin identified for it, allowing it to apply an algorithm it has knowledge about to the problem.
>
> If you would like us to add some of this discussion to the paper, let us know and we will do so.
>
> > What key insights does this benchmark provide regarding the limitations of current LLMs in in-context learning? And what potential directions could address these limitations?
>
> The main contribution of this work towards understanding the limitations of LLM ICL is demonstrating these limitations in a setting where several existing rationales given for poor LLM performance do not apply. Firstly, these are language tasks, so the explanations related to LLMs being primarily language models that have degraded performance on non-language tasks do not apply. Secondly, we compare LLMs to n-gram baselines, which do not have the capacity for world modeling. Underperforming these baselines indicates that any limitations are not the result of limits to world modeling ability. In both of these ways, I think this work demonstrates that the limitations of LLMs for ICL are fairly strong in entirely novel tasks, and cannot be easily categorized into pre-existing buckets.

---

### Note · Authors · 2026-01-16

**Comment:**

Withdrawing to enable submission to another publication.

**Withdrawal Confirmation:**

I have read and agree with the venue's withdrawal policy on behalf of myself and my co-authors.